# Practical Differentially Private Hyperparameter Tuning with Subsampling

**Antti Koskela**
Nokia Bell Labs
antti.h.koskela@nokia-bell-labs.com

**Tejas Kulkarni**
Nokia Bell Labs
tejas.kulkarni@nokia-bell-labs.com

## Abstract

Tuning the hyperparameters of differentially private (DP) machine learning (ML) algorithms often requires use of sensitive data and this may leak private information via hyperparameter values. Recently, Papernot and Steinke (2022) proposed a certain class of DP hyperparameter tuning algorithms, where the number of random search samples is randomized. Commonly, these algorithms still considerably increase the DP privacy parameter $\varepsilon$ over non-tuned DP ML model training and can be computationally heavy as evaluating each hyperparameter candidate requires a new training run. We focus on lowering both the DP bounds and the compute cost of these methods by using only a random subset of the sensitive data for the hyperparameter tuning and by extrapolating the optimal values to a larger dataset. We provide a Rényi differential privacy analysis for the proposed method and experimentally show that it consistently leads to better privacy-utility trade-off than the baseline method by Papernot and Steinke.

## 1 Introduction

Our aim is two-fold: to decrease the computational cost as well as the privacy cost of hyperparameter tuning of DP ML models. The reasons for this are clear. As the dataset sizes grow and models get more complex, blackbox optimization of hyperparameters becomes more expensive since evaluation of a single set of hyperparameters often requires retraining a new model. On the other hand, tuning the hyperparameters often depends on the use of sensitive data, so it requires privacy protection as well, as illustrated by the example by Papernot and Steinke (2022). Intuitively, the leakage from hyperparameters is much smaller than from the model parameters, however, providing tuning algorithms with low additional DP cost has turned out challenging. Current best algorithms (Papernot and Steinke, 2022) still come with a considerable DP cost overhead.

Although our methods and results are applicable to general DP mechanisms, we focus in particular on tuning of the DP stochastic gradient descent (DP-SGD) (Song et al., 2013; Bassily et al., 2014; Abadi et al., 2016) which has become the most widely used method to train ML models with DP guarantees. Compared to plain SGD, DP brings additional hyperparameters to tune: the noise level $\sigma$ and the clipping constant $C$. Additionally, also the subsampling ratio $\gamma$ affects the DP guarantees, as well as length of the training. Tuning all the hyperparameters of DP-SGD commonly requires use of sensitive data.

We use the results by Papernot and Steinke (2022) as building blocks of our methods. Their work was based on the analysis of Liu and Talwar (2019) who provided the first results for DP black-box optimization of hyperparameters, where, if the base training algorithm is $(\varepsilon, 0)$-DP, then the tuned model is approximately $(3\varepsilon, 0)$-DP. Papernot and Steinke (2022) provided a Rényi differential privacy (RDP) analysis for a class of black-box tuning algorithms, where the number of runs in the hyperparameter tuning is randomized. As the privacy bounds are in terms of RDP and assume only RDP bounds about the candidate model training algorithms, they are particularly suitable to tuning

37th Conference on Neural Information Processing Systems (NeurIPS 2023).

DP-SGD. However, still, running these algorithms increase the $\varepsilon$-values two or three-fold or more, and they can be computationally heavy as evaluating each candidate model requires training a new model. Our novelty is to consider using only a random subset of the sensitive data for the tuning part and use the output hyperparameter values (and potentially the model) for training subsequent models. Using a random subset for the privacy and computation costly part automatically leads to both lower DP privacy leakage as well as computational cost. We also consider ways to appropriately extrapolate the optimal value from the small subset of data to a larger dataset.

The RDP bounds for the DP tuning methods by Papernot and Steinke (2022) assume that the RDP-values of the candidate model training algorithms are fixed. We also consider ways to use these bounds for tuning hyperparameters that affect the RDP-values of the base algorithm, being the noise level $\sigma$, the subsampling ratio $\gamma$ and the length of training in case of DP-SGD.

## 1.1 Related Work on Hyperparameter Tuning

Chaudhuri and Vinterbo (2013) were the first ones to focus on DP bounds for hyperparameter tuning. An improvement was made by Liu and Talwar (2019) who considered black-box tuning of $(\varepsilon, \delta)$-DP mechanisms. Mohapatra et al. (2022) showed that for reasonable numbers of adaptively chosen private candidates a naive RDP accounting (i.e., RDP parameters grow linearly w.r.t. the number of model evaluations) often leads to lower DP bounds than the methods by Liu and Talwar (2019). Papernot and Steinke (2022) gave RDP bounds for black-box tuning algorithms that grow only logarithmically w.r.t. the number of model evaluations. In a non-DP setting, hyperparameter tuning with random subsamples has been considered for SVMs (Horváth et al., 2017) and for large datasets in healthcare (Waring et al., 2020). Small random subsets of data have been used in Bayesian optimization of hyperparameters (Swersky et al., 2013; Klein et al., 2017). Recent works (Killamsetty et al., 2021, 2022) consider using subsets of data for hyperparameter tuning of deep learning models.

## 1.2 Our Contributions

- We propose a subsampling strategy to lower the privacy cost and computational cost of DP hyperparameter tuning. We provide a tailored RDP analysis for the proposed strategy. Our analysis is in terms of RDP and we use existing results for tuning Papernot and Steinke (2022) and DP-SGD (Zhu and Wang, 2019) as building blocks.

- We propose algorithms to tune hyperparameters that affect the RDP guarantees of the base model training algorithms. We provide a rigorous RDP analysis for these algorithms.

- We carry out experiments on several standard datasets, where we are able to improve upon the baseline tuning method by a clear margin. While our experiments focus mainly on training of deep learning models with DP-SGD and DP-Adam, our framework is currently applicable to any computation that involves selecting the best among several alternatives (consider e.g., DP model selection, Thakurta and Smith, 2013).

## 2 Background: DP, DP-SGD and DP Hyperparameter Tuning

We first give the basic definitions. An input dataset containing $n$ data points is denoted as $X = \{x_1, \ldots, x_n\}$. Denote the set of all possible datasets by $\mathcal{X}$. We say $X$ and $Y$ are neighbors if we get one by adding or removing one data element to or from the other (denoted $X \sim Y$). Consider a randomized mechanism $\mathcal{M} : \mathcal{X} \to \mathcal{O}$, where $\mathcal{O}$ denotes the output space. The $(\varepsilon, \delta)$-definition of DP can be given as follows (Dwork, 2006).

**Definition 1.** *Let $\varepsilon > 0$ and $\delta \in [0, 1]$. We say that a mechanism $\mathcal{M}$ is $(\varepsilon, \delta)$-DP, if for all neighboring datasets $X$ and $Y$ and for every measurable set $E \subset \mathcal{O}$ we have:*

$$\Pr(\mathcal{M}(X) \in E) \le e^{\varepsilon}\Pr(\mathcal{M}(Y) \in E) + \delta.$$

We will also use the Rényi differential privacy (RDP) (Mironov, 2017) which is defined as follows. Rényi divergence of order $\alpha > 1$ between two distributions $P$ and $Q$ is defined as

$$D_\alpha(P||Q) = \frac{1}{\alpha - 1} \log \int \left( \frac{P(t)}{Q(t)} \right)^{\alpha} Q(t) \, \mathrm{d}t. \tag{2.1}$$

**Definition 2.** *We say that a mechanism $\mathcal{M}$ is $(\alpha, \varepsilon)$-RDP, if for all neighboring datasets $X$ and $Y$, the output distributions of $\mathcal{M}(X)$ and $\mathcal{M}(Y)$ have Rényi divergence of order $\alpha$ at most $\varepsilon$, i.e.,*

$$\max_{X \sim Y} D_\alpha\big(\mathcal{M}(X)||\mathcal{M}(Y)\big) \leq \varepsilon.$$

We can convert from Rényi DP to approximate DP using, for example, the following formula:

**Lemma 3** (Canonne et al. 2020). *Suppose the mechanism $\mathcal{M}$ is $(\alpha, \varepsilon')$-RDP. Then $\mathcal{M}$ is also $(\varepsilon, \delta(\varepsilon))$-DP for arbitrary $\varepsilon \geq 0$ with*

$$\delta(\varepsilon) = \frac{\exp\big((\alpha - 1)(\varepsilon' - \varepsilon)\big)}{\alpha}\left(1 - \frac{1}{\alpha}\right)^{\alpha-1}. \tag{2.2}$$

As is common in practice, we carry out the RDP accounting such that we do bookkeeping of total $\varepsilon(\alpha)$-values for a list of RDP-orders (e.g. integer $\alpha$'s) and in the end convert to $(\varepsilon, \delta)$-guarantees by minimizing over the values given by Equation(2.2) w.r.t. $\alpha$. RDP accounting for compositions of DP mechanisms is carried using standard RDP composition results (Mironov, 2017).

DP-SGD differs from SGD such that sample-wise gradients of a random mini-batch are clipped to have an $L_2$-norm at most $C$ and normally distributed noise with variance $\sigma^2$ is added to the sum of the gradients of the mini-batch (Abadi et al., 2016). One iteration is given by

$$\theta_{j+1} = \theta_j - \eta_j\Big(\frac{1}{|B|}\sum_{x \in B_j} \text{clip}(\nabla f(x, \theta_j), C) + Z_j\Big), \tag{2.3}$$

where the noise $Z_j \sim \mathcal{N}(\mathbf{0}, \frac{C^2\sigma^2}{|B|^2}I_d)$, $f$ denotes the loss function, $\theta$ the model parameters, $\eta_j$ the learning rate hyperparameter at iteration $j$ and $|B|$ is the expected batch size (if we carry out Poisson subsampling of mini-batches, $|B_j|$ varies). There are several results that enable the RDP analysis of DP-SGD iterations (Abadi et al., 2016; Balle et al., 2018; Zhu and Wang, 2019). The following result by Zhu and Wang (2019) is directly applicable to analyzing DP-SGD, however, we also use it for analyzing a variant of our hyperparameter tuning method.

**Theorem 4** (Zhu and Wang 2019). *Suppose $\mathcal{M}$ is a $(\alpha, \varepsilon(\alpha))$-RDP mechanism, w.r.t. to the add/remove neighbourhood relation. Consider the subsampled mechanism $(\mathcal{M} \circ \text{subsample}_{\text{Poisson}(\gamma)})(X)$, where $\text{subsample}_{\text{Poisson}(\gamma)}$ denotes Poisson subsampling with sampling ratio $\gamma$. If $\mathcal{M}$ is $(\alpha, \varepsilon(\alpha))$-RDP then $\mathcal{M} \circ \text{subsample}_{\text{Poisson}(\gamma)}$ is $(\alpha, \varepsilon'(\alpha))$-RDP ($\alpha \geq 2$ is an integer), where*

$$\varepsilon'(\alpha) = \frac{1}{\alpha - 1}\log\left((1-\gamma)^{\alpha-1}(\alpha\gamma - \gamma + 1) + \binom{\alpha}{2}\gamma^2(1-\gamma)^{\alpha-2}\exp(\varepsilon(2))\right.$$
$$\left. + 3\sum_{j=3}^{\alpha}\binom{\alpha}{j}\gamma^j(1-\gamma)^{\alpha-j}\exp((j-1)\varepsilon(j))\right).$$

We remark that the recent works (Koskela et al., 2020; Gopi et al., 2021; Zhu et al., 2022) give methods to carry out $(\varepsilon, \delta)$-analysis of DP-SGD tightly. As the state-of-the-art bounds for hyperparameter tuning methods are RDP bounds (Papernot and Steinke, 2022), for simplicity, we will also analyze DP-SGD using RDP.

Intuitively, the leakage from hyperparameters is much smaller than from the model parameters, however, considering it in the final accounting is needed to ensure rigorous DP guarantees. Currently the most practical $(\varepsilon, \delta)$-guarantees for DP hyperparameter tuning algorithms are those of (Papernot and Steinke, 2022). In the results of Papernot and Steinke (2022) it is important that the number of runs $K$ with the hyperparameter tuning is randomized. They analyze various distributions for drawing $K$, however, we focus on using the Poisson distribution as it is the most concentrated around the mean among all the alternatives. The corresponding hyperparameter tuning algorithm and its privacy guarantees are given by Thm.5.

First recall: $K$ is distributed according to a Poisson distribution with mean $\mu > 0$, if for all non-negative integer values $k$: $\mathbb{P}(K = k) = \mathrm{e}^{-\mu} \cdot \frac{\mu^k}{k!}$.

**Theorem 5** (Papernot and Steinke 2022)**.** *Let $Q : \mathcal{X}^N \to \mathcal{Y}$ be a randomized algorithm satisfying $\big(\alpha, \varepsilon(\alpha)\big)$-RDP and $(\widehat{\varepsilon}, \widehat{\delta})$-DP for some $\alpha \in (1, \infty)$ and $\varepsilon, \widehat{\varepsilon}, \widehat{\delta} \geq 0$. Assume $\mathcal{Y}$ is totally ordered. Let the Poisson distribution parameter $\mu > 0$. Define the hyperparameter tuning algorithm $A : \mathcal{X}^N \to \mathcal{Y}$ as follows. Draw $K$ from a Poisson distribution with mean $\mu$. Run $Q(X)$ for $K$ times. Then $A(X)$ returns the best value of those $K$ runs (both the hyperparameters and the model parameters). If $K = 0$, $A(X)$ returns some arbitrary output. If $\mathrm{e}^{\widehat{\varepsilon}} \leq 1 + \frac{1}{\alpha - 1}$, then $A$ satisfies $\big(\alpha, \varepsilon'(\alpha)\big)$-RDP, where $\varepsilon'(\alpha) = \varepsilon(\alpha) + \mu \cdot \widehat{\delta} + \frac{\log \mu}{\alpha - 1}$.*

## 3 DP Hyperparameter Tuning with a Random Subset

We next consider our main tool: we carry out the private hyperparameter tuning on a random subset, and if needed, extrapolate the found hyperparameter values to larger datasets that we use for training subsequent models. In our approach the subset of data used for tuning is generally smaller than the data used for training the final model and thus we extrapolate the hyperparameter values.

### 3.1 Our Method: Small Random Subset for Tuning

Our method works as below:

1. Use Poisson subsampling to draw $X_1 \subset X$: draw a random subset $X_1$ such that each $x \in X$ is included in $X_1$ with probability $q$.
2. Compute $(\theta_1, t_1) = \mathcal{M}_{\text{tune}}(X_1)$, where $\mathcal{M}_{\text{tune}}$ is a hyperparameter tuning algorithm (e.g., the method by Papernot and Steinke, 2022) that outputs the vector of optimal hyperparameters $t_1$ and the corresponding model parameters $\theta_1$.
3. If needed, extrapolate the hyperparameters $t_1$ to the dataset $X \setminus X_1$: $t_1 \to t_2$.
4. Compute $\theta_2 = \mathcal{M}_{\text{base}}(\theta_1, t_2, X \setminus X_1)$, where $\mathcal{M}_{\text{base}}$ is the base mechanism (e.g., DP-SGD).

Denote the whole mechanism by $\mathcal{M}$. Then, we may write

$$\mathcal{M}(X) = \big(\mathcal{M}_{\text{tune}}(X_1), \mathcal{M}_{\text{base}}\big(\mathcal{M}_{\text{tune}}(X_1), X \setminus X_1\big)\big), \tag{3.1}$$

where $X_1 \sim \text{subsample}_{\text{Poisson}(q)}(X)$. Additionally, we consider a variation of our method in which we use the full dataset $X$ instead of $X \setminus X_1$ from step 3 onwards, i.e., the mechanism

$$\mathcal{M}(X) = \big(\mathcal{M}_{\text{tune}}(X_1), \mathcal{M}_{\text{base}}\big(\mathcal{M}_{\text{tune}}(X_1), X\big)\big), \tag{3.2}$$

where $X_1 \sim \text{subsample}_{\text{Poisson}(q)}(X)$. We call these methods variant 1 and variant 2, respectively. The RDP bounds for the variant 2 can be obtained with a standard subsampling and composition result (e.g., Thm 4). We provide a tailored privacy analysis of the variant 1 in Section 3.3.

### 3.2 Extrapolating the DP-SGD Hyperparameters

We use simple heuristics to transfer the optimal hyperparameter values found for the small subset of data to a larger dataset. The clipping constant $C$, the noise level $\sigma$, the subsampling ratio $\gamma$ and the total number of iterations $T$ are kept constant in this transfer. As a consequence the $(\varepsilon, \delta)$-privacy guarantees are also the same for the models trained with the smaller and the larger dataset. For scaling the learning rate, we use the heuristics used by van der Veen et al. (2018): we scale the learning rate $\eta$ with the dataset size. I.e., if we carry out the hyperparameter tuning using a subset of size $m$ and find an optimal value $\eta^*$, we multiply $\eta^*$ by $n/m$ when transferring to the dataset of size $n$.

This can be also heuristically motivated as follows. Consider $T$ iterations of the DP-SGD (2.3). With the above rules, the distribution of the noise that gets injected into the model trained with dataset of size $n$ is

$$\sum_{j=1}^{T} Z_j \sim \mathcal{N}\left(0, \frac{T \cdot \left(\frac{n}{m}\eta^*\right)^2 \sigma^2 C^2}{(\gamma \cdot n)^2} I_d\right) \sim \mathcal{N}\left(0, \frac{T \cdot \eta^{*2} \sigma^2 C^2}{(\gamma \cdot m)^2} I_d\right)$$

which is exactly the distribution of the noise that was added to the model trained with the subsample of size $m$. This principle of keeping the noise constant when scaling the hyperparameters was also used by Sander et al. (2022).

We arrive at our scaling rule also by taking a variational Bayesian view of DP-SGD. Mandt et al. (2017) model the stochasticity of the SGD mini-batch gradients in a region approximated by a constant quadratic convex loss by invoking the central limit theorem, and arrive at a continuous-time multivariate Ornstein-Uhlenbeck (OU) process for which the discrete approximation is given by

$$\Delta\theta = -\eta g(\theta) + \frac{\eta}{\sqrt{|B|}} L \Delta W, \quad \Delta W \sim \mathcal{N}(0, I_d), \tag{3.3}$$

where $|B|$ denotes the batch size of the SGD approximation, $g(\theta)$ the full gradient and $LL^T$ the covariance matrix of the SGD noise. By minimizing the Kullback–Leibler divergence between the stationary distribution of this OU-process and the Gaussian posterior distribution $f(\theta) \propto \exp\left(-n\cdot\mathcal{L}(\theta)\right)$, where $\mathcal{L}(\theta)$ denotes the quadratic loss function and $n$ is the size of the dataset, they arrive at the expression

$$\eta^* = 2\frac{|B|}{n}\frac{d}{\mathrm{Tr}(LL^T)}$$

for the optimal learning rate value (see Thm. 1, Mandt et al., 2017). We consider the case where the additive DP noise dominates the SGD noise, and instead of the update (3.3) consider the update

$$\Delta\theta = -\eta g(\theta) + \frac{\eta \cdot \sigma \cdot C}{|B|} \Delta W, \quad \Delta W \sim \mathcal{N}(0, I_d) \tag{3.4}$$

which equals the DP-SGD update (2.3) with the mini-batch gradient replaced by the full gradient. Essentially the difference between (3.3) and (3.4) is $\sqrt{|B|}$ replaced by $|B|$, and by the reasoning used in (Thm. 1, Mandt et al., 2017), we see that the learning rate value that minimizes the KL divergence between the approximate posterior and the Gaussian posterior $f(\theta)$ is then given by

$$\eta^* = 2\frac{|B|^2}{n}\frac{d}{\mathrm{Tr}(\sigma^2 C^2 I)} = 2\frac{|B|^2}{n \cdot \sigma^2 C^2} = 2\frac{\gamma^2 n}{\sigma^2 C^2}. \tag{3.5}$$

The scaling rule (3.5) also indicates that the optimal value of the learning rate should be scaled linearly with the size of the dataset in case $\gamma$, $\sigma$ and $C$ are kept constant.

Training of certain models benefits from use of adaptive optimizers such as Adam (Kingma and Ba, 2014) or RMSProp, e.g., due to sparse gradients. Then the above extrapolation rules for DP-SGD are not necessarily meaningful anymore. In our experiments, when training a neural network classifier using Adam with DP-SGD gradients, we found that keeping the value of the learning rate fixed in the transfer to the larger dataset lead to better results than increasing it as in case of DP-SGD. We mention that there are principled ways of extrapolating the hyperparameters in non-DP setting such as those of Klein et al. (2017).

## 3.3 Privacy Analysis

The RDP analysis of the variant 2 given in Equation (3.2) is straightforward. Since the tuning set $X_1$ is sampled with Poisson subsampling with subsampling ratio $q$, we may write the mechanism as an adaptive composition

$$\mathcal{M}(X) = \left(\widetilde{\mathcal{M}}_{\mathrm{tune}}(X), \mathcal{M}_{\mathrm{base}}\big(\widetilde{\mathcal{M}}_{\mathrm{tune}}(X), X\big)\right),$$

where $\widetilde{\mathcal{M}}_{\mathrm{tune}}(X) = (\mathcal{M}_{\mathrm{tune}} \circ \mathrm{subsample}_{\mathrm{Poisson}(q)})(X)$. Using the RDP values given by Thm. 5 for $\mathcal{M}_{\mathrm{tune}}$ and the subsampling amplification result of Thm. 4, we obtain RDP bounds for $\widetilde{\mathcal{M}}_{\mathrm{tune}}(X)$. Using RDP bounds for $\mathcal{M}_{\mathrm{base}}$ (e.g., DP-SGD) and RDP composition results, we further get RDP bounds for the mechanism $\mathcal{M}$ given in (3.2).

**Tailored RDP-Analysis.** When we use the variant (3.1), i.e., we only use the rest of the data $X\backslash X_1$ for $\mathcal{M}_{\mathrm{base}}$, we can get even tighter RDP bounds. The following theorem gives tailored RDP bounds for the mechanism (3.1). Similarly to the analysis by Zhu and Wang (2019) for the Poisson subsampled Gaussian mechanism, we obtain RDP bounds using the RDP bounds of the non-subsampled mechanisms and by using binomial expansions (the proof is given in Appendix C).

**Theorem 6.** *Let $\mathcal{M}$ be the mechanism (3.1), such that the subset $X_1$ is Poisson sampled with subsampling ratio $q$, $0 \leq q \leq 1$ and let $\alpha > 1$. Denote by $\varepsilon_{\mathrm{tune}}(\alpha)$ and $\varepsilon_{\mathrm{base}}(\alpha)$ the RDP-values of mechanisms $\mathcal{M}_{\mathrm{tune}}$ and $\mathcal{M}_{\mathrm{base}}$, respectively. Then, $\mathcal{M}$ is $\big(\alpha, \varepsilon(\alpha)\big)$-RDP for*

$$\varepsilon(\alpha) = \max\{\varepsilon_1(\alpha), \varepsilon_2(\alpha)\},$$

*where*

$$\varepsilon_1(\alpha) = \frac{1}{\alpha - 1} \log \left( q^\alpha \cdot \exp \left( (\alpha - 1)\varepsilon_{\text{tune}}(\alpha) \right) + (1 - q)^\alpha \cdot \exp \left( (\alpha - 1)\varepsilon_{\text{base}}(\alpha) \right) \right.$$
$$\left. + \sum_{j=1}^{\alpha-1} \binom{\alpha}{j} \cdot q^{\alpha-j} \cdot (1-q)^j \cdot \exp \left( (\alpha - j - 1)\varepsilon_{\text{tune}}(\alpha - j) \right) \exp \left( (j - 1)\varepsilon_{\text{base}}(j) \right) \right) \tag{3.6}$$

*and*

$$\varepsilon_2(\alpha) = \frac{1}{\alpha - 1} \log \left( (1 - q)^{\alpha-1} \cdot \exp \left( (\alpha - 1)\varepsilon_{\text{base}}(\alpha) \right) \right.$$
$$\left. + \sum_{j=1}^{\alpha-1} \binom{\alpha - 1}{j} \cdot q^j \cdot (1 - q)^{\alpha-1-j} \cdot \exp \left( j \cdot \varepsilon_{\text{tune}}(j + 1) \right) \cdot \exp \left( (\alpha - j - 1)\varepsilon_{\text{base}}(\alpha - j) \right) \right). \tag{3.7}$$

**Remark 7.** *The RDP bound given by Thm. 6 is optimal in a sense that it approaches $\varepsilon_{\text{tune}}(\alpha)$ and $\varepsilon_{\text{base}}(\alpha)$ as $q \to 1$ and $q \to 0$, respectively.*

**Remark 8.** *We can initialize the subsequent model training $\mathcal{M}_{\text{base}}$ using the model $\theta_1$. This adaptivity is included in all the RDP analyses of both mechanisms (3.1) and (3.2).*

Notice that in the bounds (3.6) and (3.7) the RDP parameter of the tuning algorithm, $\varepsilon_{\text{tune}}(\alpha)$, is weighted with the parameter $q$ and the RDP parameter of the base algorithm, $\varepsilon_{\text{base}}(\alpha)$, is weighted with $1 - q$. This lowers the overall privacy cost in case the tuning set is chosen small enough.

Figure 1 illustrates how the $(\varepsilon, \delta)$-bounds of the two variants (3.1) and (3.2) behave as functions of the sampling parameter $q$ used for sampling the tuning set $X_1$, when the base mechanism DP-SGD is run for 50 epochs with the subsampling ratio $\gamma = 0.01$ and noise level $\sigma = 2.0$. The bounds for the variant 1 given in Equation (3.1) are computed using the RDP results of Thm. 6 and the bounds for the variant 2 are computed using the subsampling amplification result of Thm. 4. The RDP bounds are converted to $(\varepsilon, \delta)$-bounds using the conversion rule of Lemma 3 with $\delta = 10^{-5}$. The fact that the bounds for the variants 1 and 2 cross when $\mu = 45$ at small values of $q$ suggests that the bounds of Thm. 6 could still be tightened.

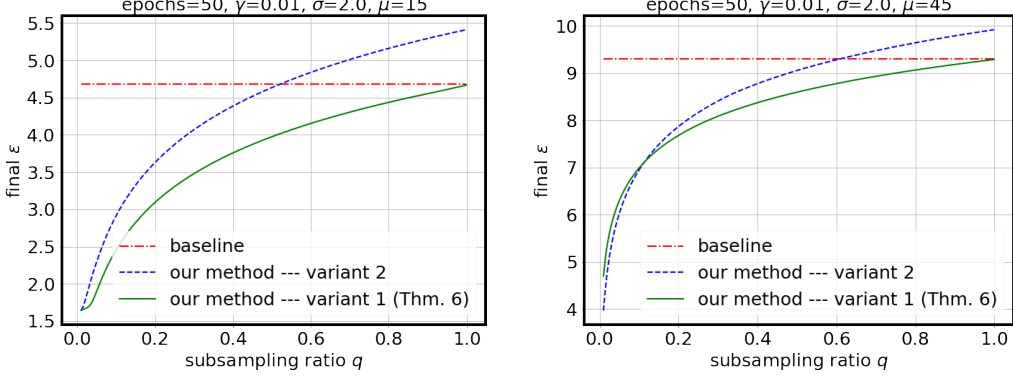

Figure 1: Comparison of $(\varepsilon, \delta)$-bounds for the variant 1 given in Equation (3.1) and the variant 2 given in Equation (3.2) as a function of the subsampling ratio $q$ used for sampling the tuning set $X_1$. Also shown is the $(\varepsilon, \delta)$-bound for the baseline algorithm described in Thm. 5. Here $\mu$ refers to the expected number of model evaluations in the tuning algorithm.

## 3.4   Computational Savings

Our scaling approach for DP-SGD described in Section 3.2 implies that the DP-SGD subsampling ratio $\gamma$, the noise level $\sigma$ and the number of iterations $T$ are the same when evaluating the private candidate models using the tuning set $X_1$ and when evaluating the final model using the larger dataset.

Thus, if we run the base algorithm for E epochs, we easily see that the expected number of required gradient evaluations for the variant 1 given in (3.1) is given by $\mathrm{E} \cdot (\mu \cdot q \cdot n + (1-q) \cdot n)$ and for the variant 2 given in (3.2) it is given by $\mathrm{E} \cdot (\mu \cdot q \cdot n + n)$, whereas the baseline requires in expectation $\mu \cdot n \cdot \mathrm{E}$ evaluations in case it is carrying out tuning with the same hyperparameter candidates as our method. Since the number of iterations $T$ is kept fixed, there are some constant overheads in the compute cost such as those coming from the model updates. Therefore the actual speed ups are slightly smaller.

For example, in our experiments with $\mu = 15$ and $q = 0.1$, the baseline requires $\frac{\mu}{\mu \cdot q + 1} \approx 6$ times more gradient evaluations than our method and when $\mu = 45$ and $q = 0.1$ the baseline requires $\approx 8$ times more gradient evaluations. The actual speed ups are shown in the figures of Section 5.

# 4 Dealing with DP-SGD Hyperparameters that Affect the DP Guarantees

Thm. 5 gives RDP-parameters of order $\alpha$ for the tuning algorithm, assuming the underlying candidate picking algorithm is $(\alpha, \varepsilon(\alpha))$-RDP. In case of DP-SGD, if we are tuning the learning rate $\eta$ or clipping constant $C$, and fix rest of the hyperparameters, these $(\alpha, \varepsilon(\alpha))$-RDP bounds are fixed for all the hyperparameter candidates. However, if we are tuning hyperparameters that affect the DP guarantees, i.e., the subsampling ratio $\gamma$, the noise level $\sigma$ or the length of the training $T$, it is less straightforward to determine suitable uniform $\varepsilon(\alpha)$-upper bounds. As is common practice, we consider a grid $\Lambda$ of $\alpha$-orders for RDP bookkeeping (e.g. integer values of $\alpha$'s).

## 4.1 Grid Search with Randomization

To deal with this problem, we first set an approximative DP target value $(\varepsilon, \delta)$ that we use to adjust some of the hyperparameters. For example, if we are tuning the subsampling ratio $\gamma$ and training length $T$, we can, for each choice of $(\gamma, T)$, adjust the noise scale $\sigma$ so that the resulting training iteration is at most $(\varepsilon, \delta)$-DP. Vice versa, we may tune $\gamma$ and $\sigma$, and take maximal value of $T$ such that the resulting training iteration is at most $(\varepsilon, \delta)$-DP.

More specifically, we first fix $\varepsilon, \delta > 0$ which represent the target approximative DP bound for each candidate model. Denote by $\varepsilon(T, \delta, \gamma, \sigma)$ the $\varepsilon$-value of the subsampled Gaussian mechanism with parameter values $\gamma, \sigma$ and $T$ and for fixed $\delta$. To each pair of $(\gamma, T)$, we attach a noise scale $\sigma_{\gamma, T}$ such that it is the smallest number with which the resulting composition is $(\varepsilon, \delta)$-DP:

$$\sigma_{\gamma, T} = \min\{\sigma \in \mathbb{R}^+ \ : \ \varepsilon(T, \delta, \gamma, \sigma) \leq \varepsilon\}.$$

As the RDP values increase monotonously w.r.t. the number of compositions, it is straightforward to find $\sigma_{\gamma, T}$, e.g., using the bisection method. Alternatively, we could fix a few values of $\sigma$, and to each combination of $(\gamma, \sigma)$, attach the largest $T$ (denoted $T_{\gamma, \sigma}$) such that the target $(\varepsilon, \delta)$-guarantee holds.

We consider a finite grid $\Gamma$ of possible hyperparameter values $t$ (e.g., $t = (\gamma, \sigma, T)$, where $T$ is adjusted to $\gamma$ and $\sigma$). Then, for all $t \in \Gamma$, we compute the corresponding RDP value $\varepsilon_t(\alpha)$ for each $\alpha \in \Lambda$. Finally, for each $\alpha \in \Lambda$, we set

$$\varepsilon(\alpha) = \max_{t \in \Gamma} \varepsilon_t(\alpha).$$

Then, since for each random draw of $t$, the DP-SGD trained candidate model is $\varepsilon(\alpha)$-RDP, by Lemma 9 given below, the candidate picking algorithm $Q$ is also $\varepsilon(\alpha)$-RDP. This approach is used in the experiments of Figure 4, where we jointly tune $T$, $\gamma$ and $\eta$.

## 4.2 RDP Analysis

For completeness, in Appendix D we prove the following result which gives RDP bounds for the case we randomly draw hyperparemeters that affect the privacy guarantees of the candidate models.

**Lemma 9.** *Denote by $\beta$ the random variable of which outcomes are the hyperparameter candidates (drawing either randomly from a grid or from given distributions). Consider an algorithm Q, that first randomly picks hyperparameters $t \sim \beta$, then runs a randomized mechanism $\mathcal{M}(t, X)$. Suppose $\mathcal{M}(t, X)$ is $(\alpha, \varepsilon(\alpha))$-RDP for all t. Then, Q is $(\alpha, \varepsilon(\alpha))$-RDP.*

# 5 Experimental Results

We perform our evaluations on standard benchmark datasets for classification: CIFAR-10 (Krizhevsky and Hinton, 2009), MNIST (LeCun et al., 1998), FashionMNIST (Xiao et al., 2017) and IMDB (Maas et al., 2011). Full details of the experiments are given in Appendix A. When reporting the results, we set $\delta = 10^{-5}$ in all experiments.

**Learning rate tuning.** Figures 2 and 3 summarize the results for learning rate tuning using our methods and the baseline method by Papernot and Steinke (2022). The learning rate grid size is either 9 or 10 and the grid values are specified in Table 2 (Appendix). We fix the subsampling ratio $\gamma$ and the number of epochs to the values given in Table 1 (Appendix) for all models. The batch sizes in the submodels are defined by scaling $\gamma$ on the corresponding dataset sizes. We use $q = 0.1$ which means that, for example, if the Poisson subsampling of DP-SGD gives in expectation a batch size of 128 in the tuning phase of our methods, the expected batch sizes for the final models of variant 1 and 2 are 1152 and 1280, respectively. We use $\mu = 15$ (the expected number of runs for the tuning algorithm).

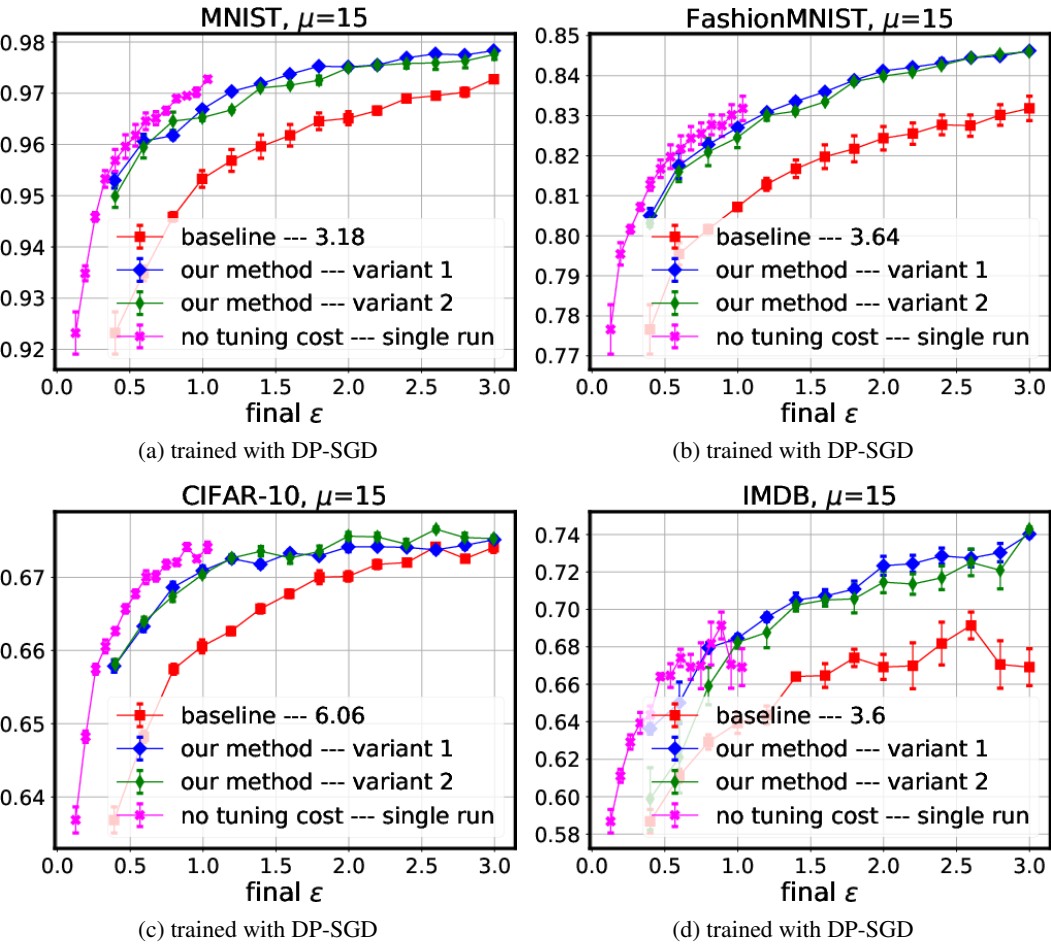

Figure 2: Tuning learning rate with DP-SGD. Test accuracies are averaged across 10 independent runs and the error bars denote the standard error of the mean. The numbers in the legends refer to the mean training timings of the baseline scaled with respect to minimum of variant 1 and 2. For example, for CIFAR-10, the average training time for the baseline method is 6.06 times bigger than for the fastest of our methods. For perspective, we also add curves showing the privacy cost of training a single model with optimal hyperparameters obtained from the baseline.

**Tuning all hyperparameters.** Next, we jointly optimize the number of epochs, the DP-SGD subsampling ratio $\gamma$, and the learning rate $\eta$ using the hyperparameter candidates given in Table 2 (Appendix). The remaining setup is the same as in the previous experiment. Figure 4 shows the same

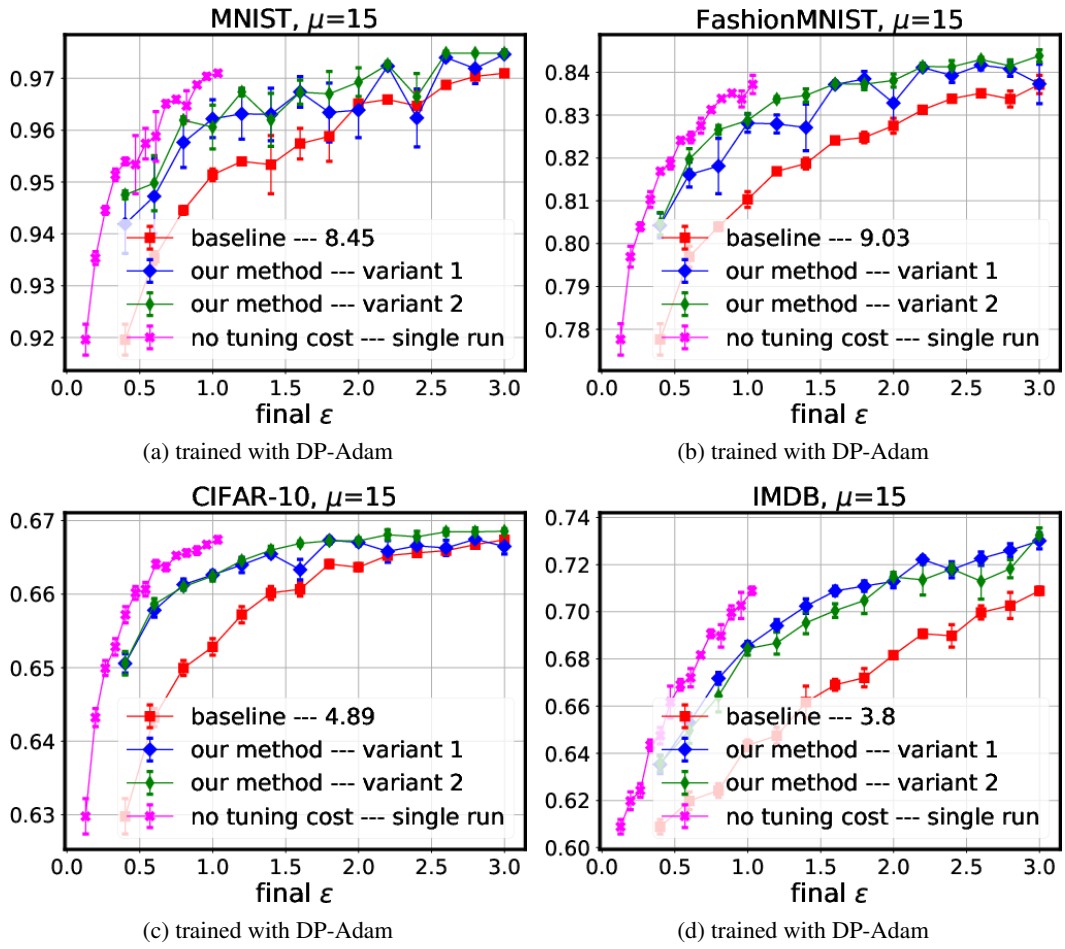

Figure 3: Tuning learning rate with DP-Adam. Test accuracies are averaged across 10 independent runs and the error bars denote the standard error of the mean. The numbers in the legends refer to the mean training timings of the baseline scaled with respect to minimum of variant 1 and 2. For example, for FashionMNIST, the average training time for the baseline method is 9.03 times bigger than the fastest of our methods. For perspective, we also add curves showing the privacy cost of training a single model with optimal hyperparameters obtained from the baseline. Figure 6 (Appendix) shows a more detailed version of this plot.

quantities as Figures 2 and 3, however, higher values of $\mu$ are used to accommodate to increased hyperparameter spaces.

**Takeaways.** Overall, for both DP-SGD and DP-Adam, we observe that both variants of our method provide better privacy-utility trade-off and have a lower computational cost than the baseline method. Additionally, we also note the benefits of tailored analysis (Thm 6) in Figure 2 in terms of slightly higher accuracy for variant 1 compared to variant 2 for DP-SGD. In Figure 4 where we are also tuning the batch size and epochs, the noise levels are higher compared to Figure 2. One reason for slightly better privacy-utility trade-off for variant 2 with DP-SGD is possibly the fact that our tailored bound is less tight for higher values of $\mu$ and small values of $q$ (see Figure 1).

## 6 Discussion

We have considered a simple strategy for lowering the privacy cost and computational cost of DP hyperparameter tuning: we carry out tuning using a random subset of data and extrapolate the optimal values to the larger training set used to train the final model. We have also provided methods to tune the hyperparameters that affect the DP guarantees of the model training themselves, those

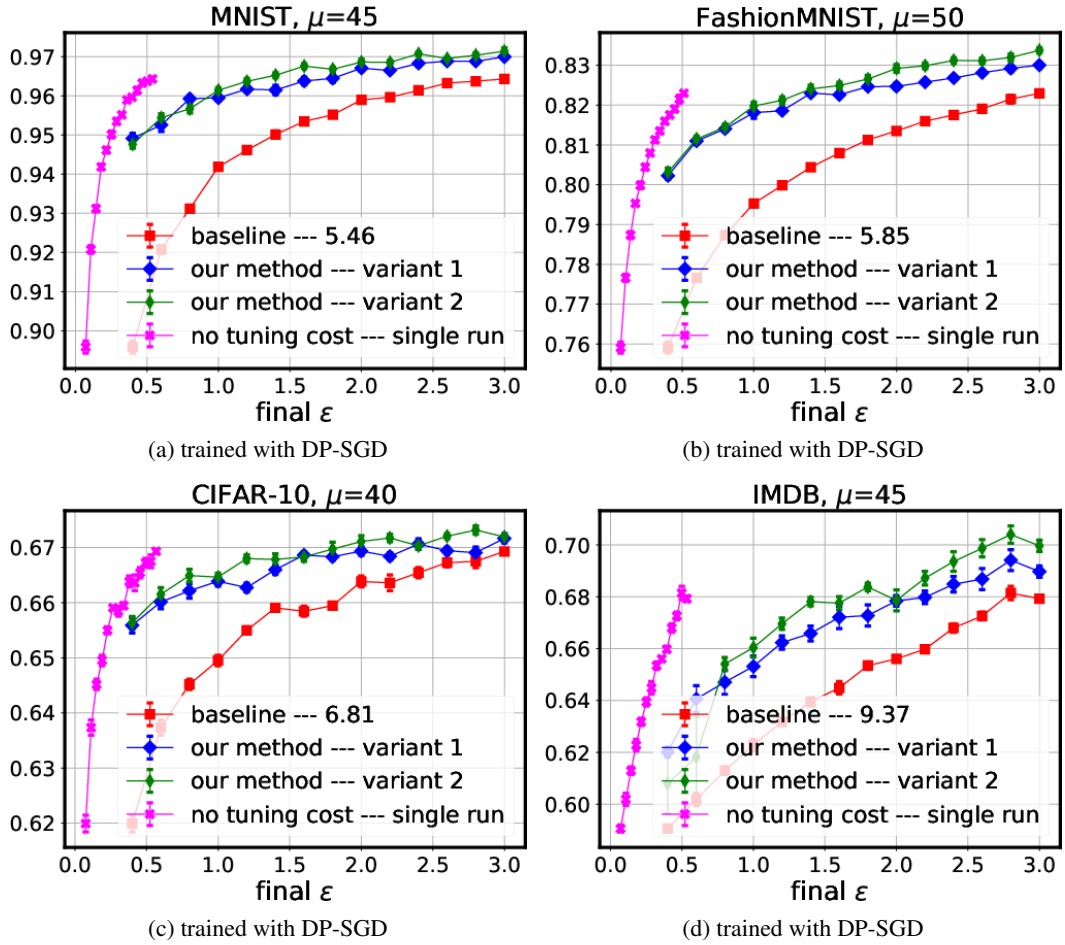

Figure 4: Tuning of subsampling ratio, training epochs, and learning rate with DP-SGD. Test accuracies are averaged across 10 independent runs and the error bars denote the standard error of the mean. The numbers in the legends refer to the mean timings of the baseline method scaled with respect to the minimum of variant 1 and 2. For perspective, we also add curves showing the privacy cost of training a single model with optimal hyperparameters obtained from the baseline. Figure 7 (Appendix) shows a more detailed version of this plot.

being the noise level and subsampling ratio in case of DP-SGD. Our experiments show a clear improvement over the baseline method by Papernot and Steinke (2022) when tuning DP-SGD for neural networks and using simple but well-justified heuristics for the extrapolation. One obvious limitation of our method is that it is limited to DP-SGD although we show also positive results for the Adam optimizer combined with DP-SGD gradients. Finding out whether effective scaling rules could be derived for DP-SGD with momentum, DP-FTRL (Kairouz et al., 2021) or more refined adaptive DP optimizers such as DP²-RMSprop (Li et al., 2023), is left for future work. An interesting avenue of future work is also to find more black-box type of extrapolation methods, something that has been considered in the non-DP case (see, e.g. Klein et al., 2017). Another interesting question is how to carry out more accurate privacy analysis using the so-called privacy loss distributions (PLDs) and numerical accounting. The main reason for using RDP instead of PLD accounting in this work was that the tightest privacy bounds for DP hyperparameter tuning are given in terms of RDP. We have used Poisson subsampling to obtain the tuning set as it is relatively easy to analyze, however, other sampling methods could be analyzed as well.

# 7 Broader Impact Statement

Our present work makes DP machine learning more appealing and potentially more widespread in practice. This can improve privacy protection in general but can also carry negative side effects. The proposed method can make DP learning more appealing by improving DP hyperparameter tuning leading to higher utility machine learning models at equivalent provable total privacy cost. Our method has lower compute cost than some alternatives, potentially leading to considerable resource savings. While DP gives strong privacy guarantees, it may have negative impacts as well, as some DP learning methods have for example been shown to have a relatively lower utility for minorities.

## Acknowledgments

We would like to thank our colleague Laith Zumot for discussions about hyperparameter tuning methods at the early stages of the project.

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

# Appendix

## A Full Description of Experiments

**Quality Metric and Evaluation**. In all of our experiments, we have a partitioning of the available data into train and test sets and we choose the best model based on the test accuracy. The quality score applied on the test set is a relatively low sensitivity function, and therefore, even for a private test set, parallel composition (for an RDP bound of parallel compositions, we refer to Appendix E) can accommodate DP evaluation of a quality metric in the training budget itself. However, we assume that only the training dataset is private and the test data is public for simplicity. This assumption of test dataset being public and the approach of making only two (train and test) partitions of available data instead of three (train, validation, and test) has been considered in many prior works (Mohapatra et al., 2022; Papernot and Steinke, 2022) to study the proposed method in isolation.

Our utility plots (Figures 2 to Figure 8) show the test accuracy of the final model against the final approximate DP $\varepsilon$ which includes the privacy cost of the tuning process and of the final model for all methods. We fix $\delta = 10^{-5}$ always when reporting the $\varepsilon$-values. We fix $q = 0.1$ for all of our methods in all experiments. We mention that in several cases smaller value of $q$ would have lead to better privacy-accuracy trade-off of the final model, however, we use the same value $q = 0.1$ in all experiments for consistency.

**Methods.** We consider the both variants of our proposed approach in our experiments. The RDP parameters for the variant 1 are obtained by using Thm. 6 (new RDP result) and for variant 2 by combining Thm. 4 (subsampling amplification) with Thm. 5 (RDP cost of the hyperparameter tuning). We scale the hyperparameters in our methods for the training data of the final model as discussed in Sec. 3.2. The method by Papernot and Steinke (2022) described in Thm. 5 is the baseline.

**Datasets and Models.** We carry out our experiments on the following standard benchmark datasets for classification: CIFAR-10 (Krizhevsky and Hinton, 2009), MNIST (LeCun et al., 1998), FashionMNIST (Xiao et al., 2017) and IMDB (Maas et al., 2011). For MNIST and IMDB, we use the convolutional neural networks from the examples provided in the Opacus library Yousefpour et al. (2021). For FashionMNIST, we consider a simple feedforward 3-layer network with hidden layers of width 120. For CIFAR-10, we use a Resnet20 pre-trained on CIFAR-100 (Krizhevsky and Hinton, 2009) dataset so that only the last fully connected layer is trained. We minimize the cross-entropy loss in all models. We optimize with DP-SGD and DP-Adam for all datasets. As suggested by Papernot et al. (2020), we replace all ReLU activations with tempered sigmoid functions in CIFAR-10, MNIST, and FashionMNIST networks, which limits the magnitudes of non-private gradients and improves model accuracies.

**Hyperparameters**. For these datasets, in one of the experiments we tune only the learning rate ($\eta$), and in the other one $\eta$, $\gamma$ (subsampling ratio), and the number of epochs, while fixing the clipping constant $C$. For DP-Adam, we do not scale the learning rate. The number of trainable parameters and the hyperparameter grids are provided in Table 2 (Appendix B). The numbers of epochs are chosen to suit our computational constraints. Following the procedure described in Section 4.1, we always adjust $\sigma$ such that we compute the smallest $\sigma$ that satisfies a target final $(\varepsilon, \delta)$ bound for each $(\gamma,$ epoch) pair. Furthermore following the RDP accounting procedure desribed in Section 4.1 we obtain uniform RDP guarantees for the candidate models and furthermore RDP guarantees for our methods.

**Implementation**. For the implementation of DP-SGD and DP-Adam, we use the Opacus library (Yousefpour et al., 2021). For scalability, we explore the hyperparameter spaces with Ray Tune (Liaw et al., 2018) on a dedicated multi-GPU cluster. We use two sets of gpus and one set is much weaker than the other. All three methods have independent randomness (e.g. $K$ hyperparameter candidates sampled, DP noise). Ray tune maintains a job queue and the number of models trained parallelly depends on the gpu count. Each model is trained on 0.5 gpu, and each gpu has enough memory to accommodate 2 models. Models finish their entire training on the same gpu that was allocated to them at the start of their training.

**Measuring training time**. We want to reduce the dependence of our measurements on the available hardware resources (e.g. number of gpus). For each method, we store the per epoch time in seconds for each model being trained in parallel, and finally sum the timings for all epochs of all $K$ models (as if they were trained serially). The clock for each model starts only when it is under execution. Among variant 1 and 2, the final training time depends mainly on the value of K sampled, though

variant 1 trains the final model with slightly smaller data. The baseline method takes much more time to run compared to our methods on weaker GPUs, which explains the differences in speed gains.

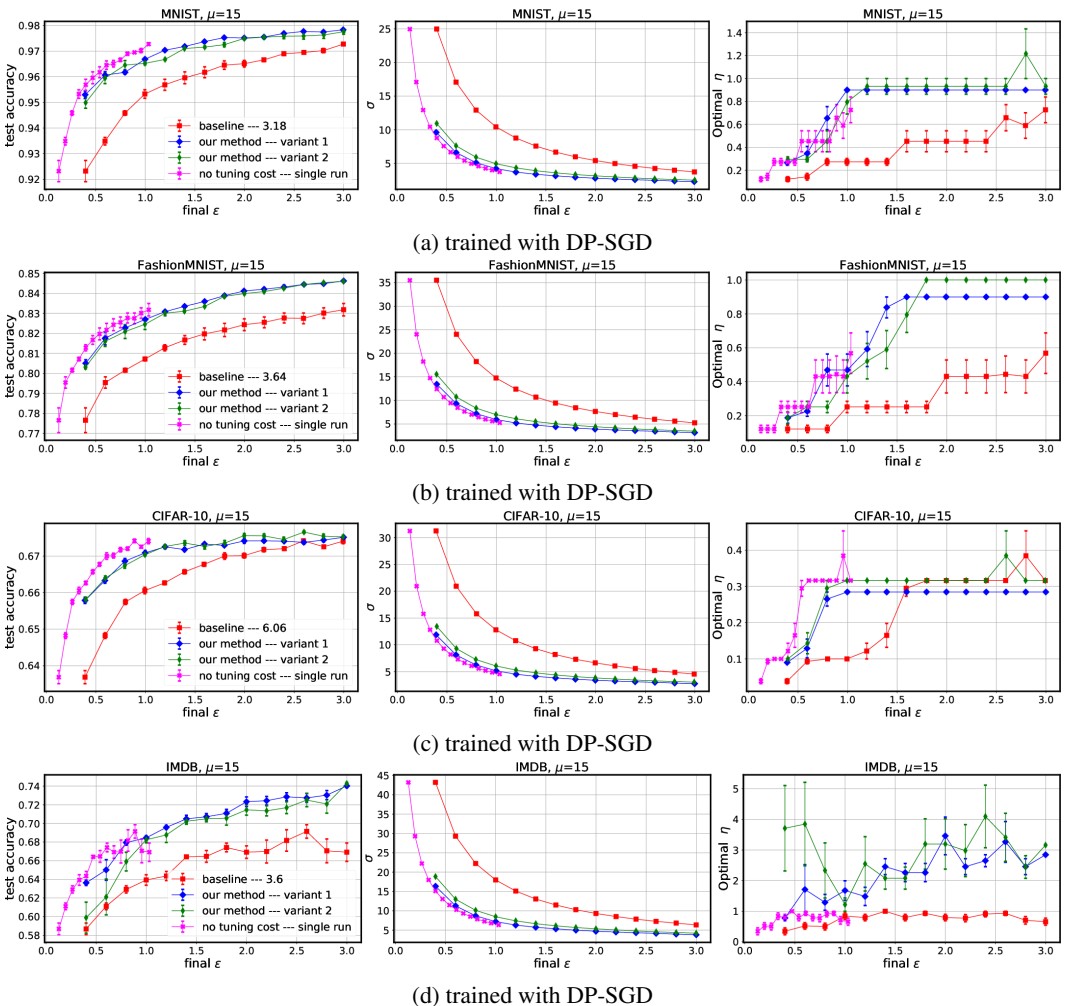

Figure 5: Tuning only the learning rate with DP-SGD. Test accuracies are averaged across 10 independent runs and the error bars denote the standard error of the mean. The number in the legends in the first column refer to the scaled mean training timings for the baseline method with respect to the fastest of variant 1 and 2. The second column plots final $\varepsilon$ vs. mean $\sigma$. Our methods inject significantly smaller noise compared to the baseline for all $\varepsilon$ regimes. We also observe that due to tight analysis in Thm 6, $\sigma$ for variant 1 is consistently lower than for variant 2. As a result, we see slightly higher accuracy for variant 1 in many cases. The third column plots final $\varepsilon$ vs. mean optimal $\eta$. Note that due to randomess in the candidate selection process, optimal $\eta$'s for all three methods need not be the same. For perspective, we also add curves showing the privacy cost of training a single model with optimal hyperparameters obtained from the baseline.

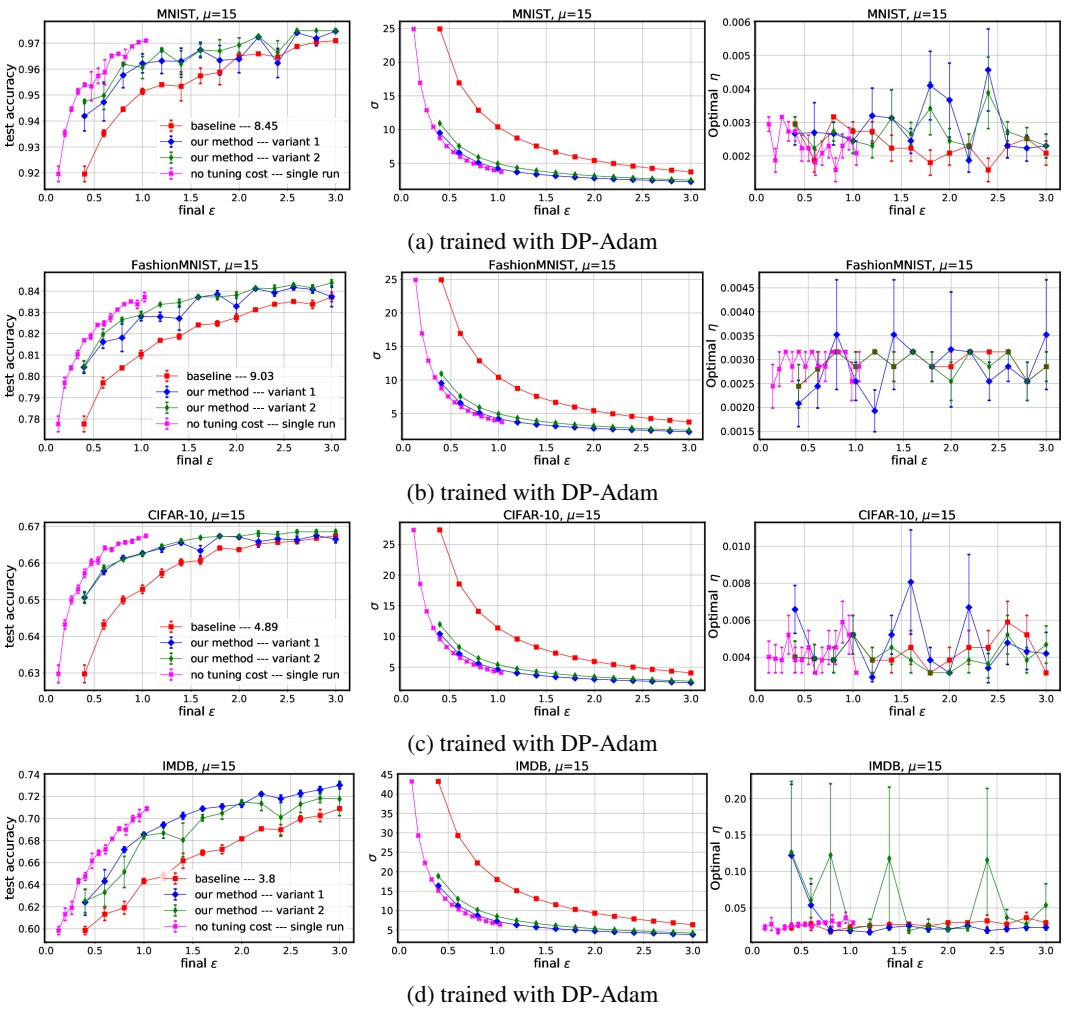

Figure 6: Tuning only the learning rate with DP-Adam. Test accuracies are averaged across 10 independent runs and the error bars denote the standard error of the mean. No learning rate scaling was applied. The number in the legends in the first column refer to the scaled mean training timings for the baseline method with respect to the fastest of variant 1 and 2. The second column plots final $\varepsilon$ vs. mean $\sigma$. Our methods inject significantly smaller noise compared to the baseline for all $\varepsilon$ regimes. We also observe that due to tight analysis in Thm 6, $\sigma$ for variant 1 is consistently lower than for variant 2. As a result, we see slightly higher accuracy for variant 1 in many cases. The third column plots final $\varepsilon$ vs. mean optimal $\eta$. Note that due to randomess in the candidate selection process, optimal $\eta$'s for all three methods need not be the same. For perspective, we also add curves showing the privacy cost of training a single model with optimal hyperparameters obtained from the baseline.

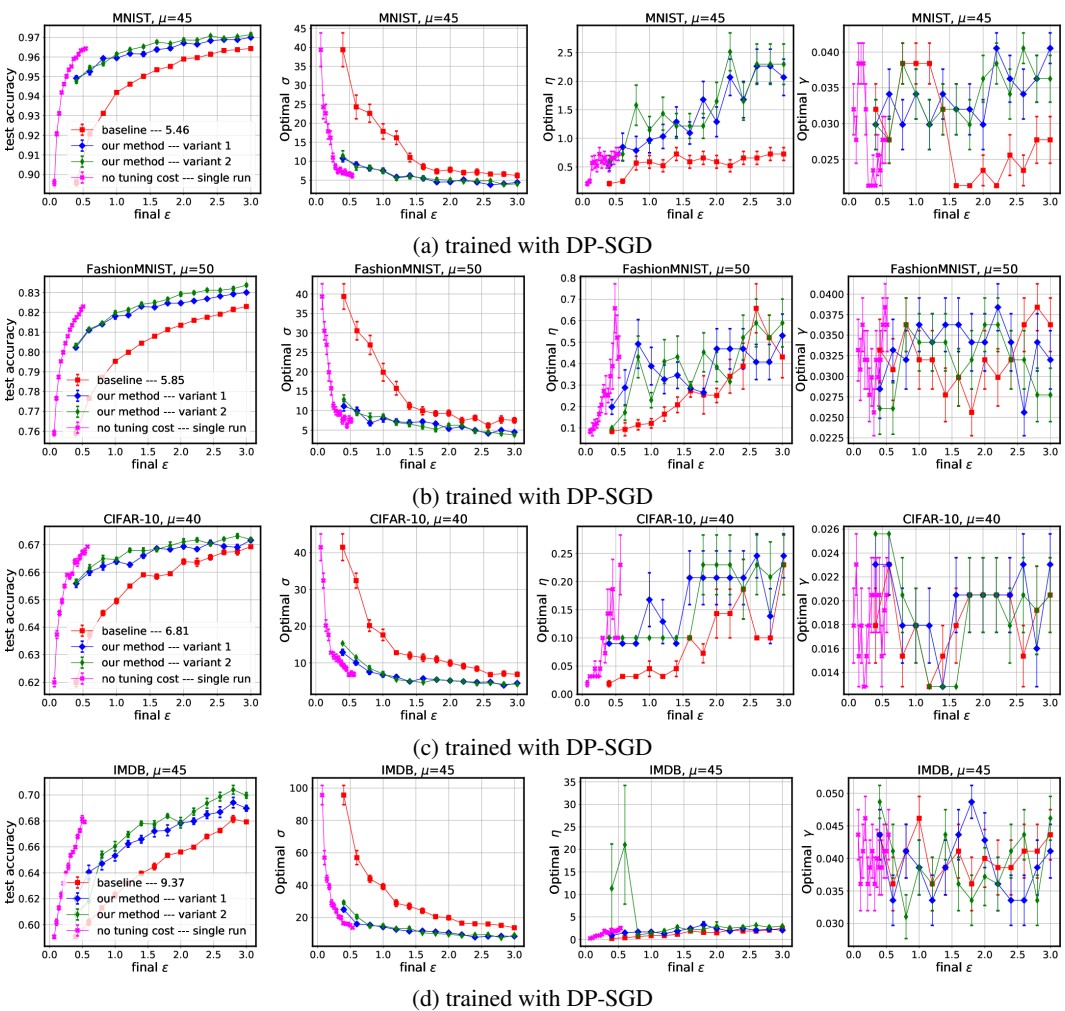

Figure 7: Tuning all hyperparameters ($\eta$, epochs, and $\gamma$) with DP-SGD. Test accuracies are averaged across 10 independent runs and the error bars denote the standard error of the mean. The numbers in the legends in the first column refer to the scaled mean training timings for the baseline method with respect to the fastest of variant 1 and 2. The second column plots final $\varepsilon$ vs. mean $\sigma$. Our methods inject significantly smaller noise compared to the baseline for all $\varepsilon$ regimes even at high values of $\mu$. Since there are several hyperparameters at play together, it is difficult to attribute slightly higher accuracy for variant 2 compared to 1 to any one factor. However, we suspect slight inferiority of our bound for higher values of $\mu$ (Figure 1, right plot) and higher training dataset for the final model in variant 2 could be the main reasons. The third (and fourth) column plots final $\varepsilon$ vs. mean optimal $\eta$ (and $\gamma$). Noise level $\sigma$ is a proxy for number of epochs, since we obtain a $\sigma$ for each combination of $\gamma$ and number of epochs. Therefore, we omit plots showing final $\epsilon$ vs. epochs plot for readability. Note that due to randomness in the candidate selection process, optimal $\eta$'s and $\gamma$'s for all three methods need not be the same. For perspective, we also add curves showing the privacy cost of training a single model with optimal hyperparameters obtained from the baseline.

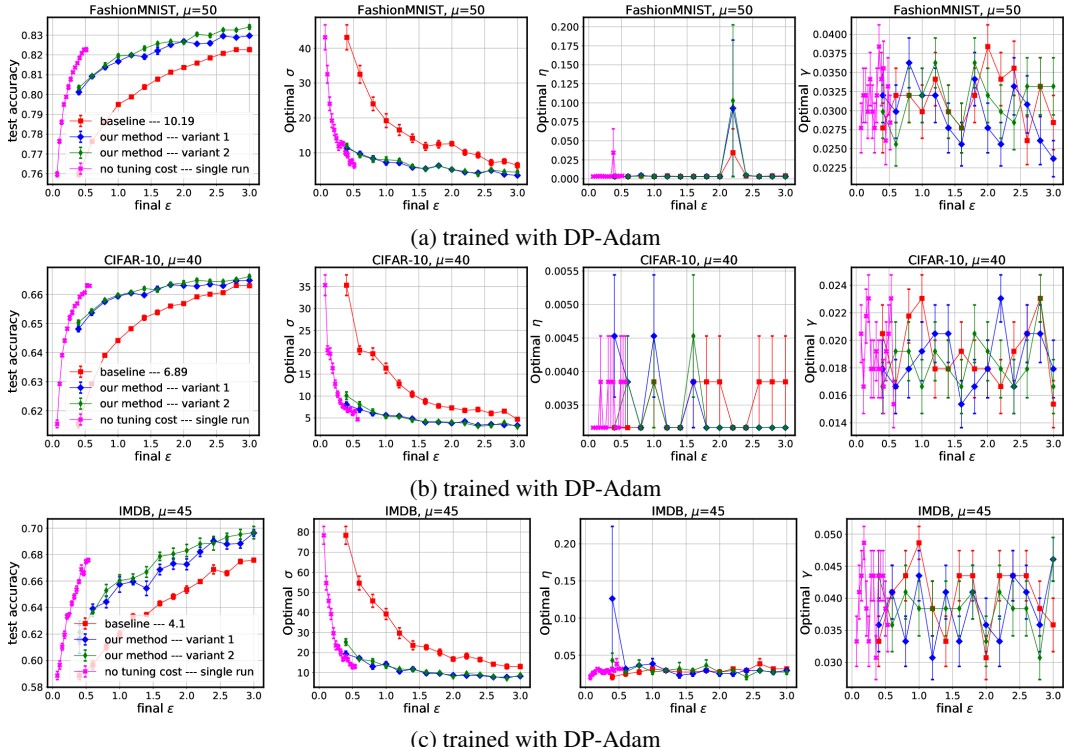

Figure 8: Tuning all hyperparameters ($\eta$, epochs, and $\gamma$) with DP-Adam. Test accuracies are averaged across 10 independent runs and the error bars denote the standard errors of the means. The number in the legends in the first column refer to the scaled mean training timings for the baseline method with respect to the fastest of variant 1 and 2. The second column plots final $\varepsilon$ vs. mean $\sigma$. Our methods inject significantly smaller noise compared to the baseline for all $\varepsilon$ regimes even at high values of $\mu$. Since there are several hyperparameters at play together, it is difficult to attribute slightly higher accuracy for variant 2 compared to 1 to any one factor. However, we suspect slight inferiority of our bound for higher values of $\mu$ (Figure 1, right plot) and higher training dataset for the final model in variant 2 could be the main reasons. The third (and fourth) column plots final $\varepsilon$ vs. mean optimal $\eta$ (and $\gamma$). Noise level $\sigma$ is a proxy for number of epochs, since we obtain a $\sigma$ for each combination of $\gamma$ and number of epochs. Therefore, we omit plots showing final $\epsilon$ vs. epochs plot for readability. Note that due to randomness in the candidate selection process, optimal $\eta$'s and $\gamma$'s for all three methods need not be the same.

# B Hyperparameter Tables

Table 1: Tuning $\eta$: rest of the hyperparameters are fixed to these values. The hyperaparameter grids for the learning rates are the same as in the second experiment and are given in Table 2.

|  | MNIST | FashionMNIST | CIFAR-10 | IMDB |
|---|---|---|---|---|
| $\gamma = \frac{B}{N}$ | 0.0213 | 0.0213 | 0.0256 | 0.0256 |
| epochs | 40 | 40 | 40 | 110 |

Table 2: Tuning $\sigma$, $\eta$ and $T$: datasets and the corresponding hyperparameter grids.

|  | train/test set | parameters | C | B | learning rate ($\eta$) | epochs |
|---|---|---|---|---|---|---|
| MNIST | 60k/10k | ~26k | 1 | $\{128, 256\}$ | $\{10^{-i}\}_{i \in \{4,3.5,3,2.5,2,1.5,1,0.5,0\}}$ | {10,20,30,40} |
| FashionMNIST | 60k/10k | ~109k | 3 | $\{128, 256\}$ | $\{10^{-i}\}_{i \in \{3,2.5,2,1.5,1,0.5,0,-0.5,-1,-1.5\}}$ | {10,20,30,40} |
| CIFAR-10 | 50k/10k | 0.65k | 3 | $\{64, 128\}$ | $\{10^{-i}\}_{i \in \{3,2.5,2,1.5,1,0.5,0,-0.5,-1,-1.5\}}$ | {20,30,40} |
| IMDB | 25k/25k | ~464k | 1 | $\{64, 128\}$ | $\{10^{-i}\}_{i \in \{4,3.5,3,2.5,2,1.5,1,0.5,0\}}$ | {70,90,110} |

# C Proof of Thm 6

*Proof.* For the proof, let us denote for short $\mathcal{M}_{\text{tune}} =: \mathcal{M}_1$ and $\mathcal{M}_{\text{base}} =: \mathcal{M}_2$. Let $X \in \mathcal{X}^n$ and $x' \in \mathcal{X}$ We first consider bounding the Rényi divergence $D_\alpha\big(\mathcal{M}(X \cup \{x'\})||\mathcal{M}(X)\big)$. Looking at our approach which uses Poisson subsampling with subsampling ratio $q$ to obtain the dataset $X_1$, and conditioning the output on the randomness in choosing $X_1$, we can write the mechanism as a mixture over all possible choices of $X_1$ as

$$\mathcal{M}(X) = \sum_{X_1} p_{X_1} \cdot \big(\mathcal{M}_1(X_1), \mathcal{M}_2(\mathcal{M}_1(X_1), X \backslash X_1)\big), \tag{C.1}$$

where $p_{X_1}$ is the probability of sampling $X_1$. Since each data element is in $X_1$ with probability $q$, we can furthermore write $\mathcal{M}(X \cup \{x'\})$ as a mixture

$$\mathcal{M}(X \cup \{x'\}) = \sum_{X_1} p_{X_1} \cdot \Big(q \cdot \big(\mathcal{M}_1(X_1 \cup \{x'\}), \mathcal{M}_2(\mathcal{M}_1(X_1 \cup \{x'\}), X \backslash X_1)\big)$$
$$+ (1-q) \cdot \big(\mathcal{M}_1(X_1), \mathcal{M}_2(\mathcal{M}_1(X_1), X \backslash X_1 \cup \{x'\})\big)\Big). \tag{C.2}$$

From the quasi-convexity of the Rényi divergence (Van Erven and Harremos, 2014) and the expressions (C.1) and (C.2), it follows that

$$D_\alpha\big(\mathcal{M}(X \cup \{x'\})||\mathcal{M}(X)\big) \leq \sup_{X_1} D_\alpha\big(q \cdot \big(\mathcal{M}_1(X_1 \cup \{x'\}), \mathcal{M}_2(\mathcal{M}_1(X_1 \cup \{x'\}), X \backslash X_1)\big)$$
$$+ (1-q) \cdot \big(\mathcal{M}_1(X_1), \mathcal{M}_2(\mathcal{M}_1(X_1), X \backslash X_1 \cup \{x'\})\big)||\big(\mathcal{M}_1(X_1), \mathcal{M}_2(\mathcal{M}_1(X_1), X \backslash X_1)\big)\big). \tag{C.3}$$

Our aim is to express the right-hand side of the inequality (C.3) in terms of RDP parameters of $\mathcal{M}_1$ and $\mathcal{M}_2$. To this end, take an arbitrary $X_1 \subset X$, and denote by

- $\widetilde{P}(t)$ the density function of $\mathcal{M}_1(X_1 \cup \{x'\})$,

- $P(t)$ the density function of $\mathcal{M}_1(X_1)$,

- $\widetilde{Q}(t, s)$ the density function of $\mathcal{M}_2(t, X \backslash X_1 \cup \{x'\})$ for auxiliary variable $t$ (the output of $\mathcal{M}_1$),

- $Q(t, s)$ the density function of $\mathcal{M}_2(t, X \backslash X_1)$ for auxiliary variable $t$.

Then, we see that

$$\mathbb{P}\big(\big(\mathcal{M}_1(X_1), \mathcal{M}_2(\mathcal{M}_1(X_1), X \backslash X_1)\big) = (t, s)\big) = P(t) \cdot Q(t, s)$$

and similarly that

$$
\begin{aligned}
\mathbb{P}&\big(q \cdot \big(\mathcal{M}_1(X_1 \cup \{x'\}), \mathcal{M}_2(\mathcal{M}_1(X_1 \cup \{x'\}), X\backslash X_1)\big) \\
&+ (1-q) \cdot \big(\mathcal{M}_1(X_1), \mathcal{M}_2(\mathcal{M}_1(X_1), X\backslash X_1 \cup \{x'\})\big) = (t,s)\big) \\
&= q \cdot \mathbb{P}\big((\mathcal{M}_1(X_1 \cup \{x'\}), \mathcal{M}_2(\mathcal{M}_1(X_1 \cup \{x'\}), X\backslash X_1)) = (t,s)\big) \\
&\quad + (1-q) \cdot \mathbb{P}\big((\mathcal{M}_1(X_1), \mathcal{M}_2(\mathcal{M}_1(X_1), X\backslash X_1 \cup \{x'\})) = (t,s)\big) \\
&= q \cdot \widetilde{P}(t) \cdot Q(t,s) + (1-q) \cdot P(t) \cdot \widetilde{Q}(t,s).
\end{aligned}
$$

By the definition of the Rényi divergence, we have that

$$
\begin{aligned}
\exp\bigg( &(\alpha - 1)D_\alpha\big(q \cdot \big(\mathcal{M}_1(X_1 \cup \{x'\}), \mathcal{M}_2(\mathcal{M}_1(X_1 \cup \{x'\}), X\backslash X_1)\big) \\
&+ (1-q) \cdot \big(\mathcal{M}_1(X_1), \mathcal{M}_2(\mathcal{M}_1(X_1), X\backslash X_1 \cup \{x'\}))\| (\mathcal{M}_1(X_1), \mathcal{M}_2(\mathcal{M}_1(X_1), X\backslash X_1)))\big) \bigg) \\
&= \int \int \left( \frac{q \cdot \widetilde{P}(t) \cdot Q(t,s) + (1-q) \cdot P(t) \cdot \widetilde{Q}(t,s)}{P(t) \cdot Q(t,s)} \right)^\alpha \cdot P(t) \cdot Q(t,s) \, dt \, ds.
\end{aligned}
$$

(C.4)

which can be expanded as

$$
\begin{aligned}
\int &\int \left( \frac{q \cdot \widetilde{P}(t) \cdot Q(t,s) + (1-q) \cdot P(t) \cdot \widetilde{Q}}{P(t) \cdot Q(t,s)} \right)^\alpha P(t) \cdot Q(t,s) \, dt \, ds \\
&= \int \int \left( q \cdot \frac{\widetilde{P}(t)}{P(t)} + (1-q) \cdot \frac{\widetilde{Q}(t,s)}{Q(t,s)} \right)^\alpha P(t) \cdot Q(t,s) \, dt \, ds \\
&= \int \int q^\alpha \left( \frac{\widetilde{P}(t)}{P(t)} \right)^\alpha P(t) \cdot Q(t,s) \, dt \, ds \\
&\quad + \int \int (1-q)^\alpha \left( \frac{\widetilde{Q}(t,s)}{Q(t,s)} \right)^\alpha P(t) \cdot Q(t,s) \, dt \, ds \\
&\quad + \int \int \alpha \cdot q^{\alpha-1} \cdot (1-q) \cdot \left( \frac{\widetilde{P}(t)}{P(t)} \right)^{\alpha-1} P(t) \cdot \widetilde{Q}(t,s) \, dt \, ds \\
&\quad + \int \int \alpha \cdot q \cdot (1-q)^{\alpha-1} \cdot \left( \frac{\widetilde{Q}(t,s)}{Q(t,s)} \right)^{\alpha-1} Q(t,s) \cdot \widetilde{P}(t) \, dt \, ds \\
&\quad + \int \int \sum_{j=2}^{\alpha-2} \binom{\alpha}{j} \cdot q^{\alpha-j} \cdot (1-q)^j \cdot \left[ \left( \frac{\widetilde{P}(t)}{P(t)} \right)^{\alpha-j} P(t) \right] \left[ \left( \frac{\widetilde{Q}(t,s)}{Q(t,s)} \right)^j Q(t,s) \right] dt \, ds.
\end{aligned}
$$

(C.5)

We next bound five integrals on the right hand side of Equation (C.5). For the first two integrals, we use the RDP-bounds for $\mathcal{M}_1$ and $\mathcal{M}_2$ to obtain

$$
\int \int \left( \frac{\widetilde{P}(t)}{P(t)} \right)^\alpha P(t) Q(t,s) \, dt \, ds = \int \left( \frac{\widetilde{P}(t)}{P(t)} \right)^\alpha P(t) \, dt \le \exp\big((\alpha - 1)\varepsilon_1(\alpha)\big). \qquad \text{(C.6)}
$$

and

$$
\int \int \left( \frac{\widetilde{Q}(t,s)}{Q(t,s)} \right)^\alpha Q(t,s) P(t) \, ds \, dt \le \int \exp\big((\alpha - 1)\varepsilon_2(\alpha)\big) P(t) \, dt = \exp\big((\alpha - 1)\varepsilon_2(\alpha)\big),
$$

(C.7)

where $\varepsilon_1$ and $\varepsilon_2$ give the RDP-parameters of order $\alpha$ for $\mathcal{M}_1$ and $\mathcal{M}_2$, respectively. The third and fourth integral can be bounded analogously. In the second inequality we have also used the fact

that the RDP-parameters of $\mathcal{M}_2$ are independent of the auxiliary variable $t$. Similarly, for the third integral, we have

$$
\begin{aligned}
\int \int & \left[ \left( \frac{\widetilde{P}(t)}{P(t)} \right)^{\alpha - j} P(t) \right] \left[ \left( \frac{\widetilde{Q}(t,s)}{Q(t,s)} \right)^{j} Q(t,s) \right] \, \mathrm{d}s \, \mathrm{d}t \\
\le & \int \left[ \left( \frac{\widetilde{P}(t)}{P(t)} \right)^{\alpha - j} P(t) \right] \exp \big( (j-1)\varepsilon_2(j) \big) \, \mathrm{d}t \\
\le & \exp \big( (\alpha - j - 1)\varepsilon_1(\alpha - j) \big) \cdot \exp \big( (j-1)\varepsilon_2(j) \big).
\end{aligned}
\tag{C.8}
$$

Substituting (C.6), (C.7) (and similar expressions for the third and fourth integral) and (C.8) to Equation (C.5), we get a bound for the right-hand side of Equation (C.4). Since $X_1 \subset X$ was arbitrary, we arrive at the claim via the inequality (C.3).

Next, we consider bounding $D_\alpha\big(\mathcal{M}(X)||\mathcal{M}(Y)\big)$. The proof goes similarly as the one for $D_\alpha\big(\mathcal{M}(Y)||\mathcal{M}(X)\big)$. Denote

$$
\varepsilon_2(\alpha) = D_\alpha\big(\mathcal{M}(X)||\mathcal{M}(X \cup \{x'\})\big).
$$

With the notation of proof of Thm. 6, we see that, instead of the right-hand side of (C.4), we need to bound

$$
\begin{aligned}
& \exp \big( (\alpha - 1)\varepsilon_2(\alpha) \big) \\
= & \int \int \left( \frac{P(t) \cdot Q(t,s)}{q \cdot \widetilde{P}(t) \cdot Q(t,s) + (1-q) \cdot P(t) \cdot \widetilde{Q}(t,s)} \right)^{\alpha} \big( q \cdot \widetilde{P}(t) \cdot Q(t,s) + (1-q) \cdot P(t) \cdot \widetilde{Q}(t,s) \big) \, \mathrm{d}t \, \mathrm{d}s.
\end{aligned}
$$

In order to use here the series approach, we need to use Lemma 10:

$$
\begin{aligned}
& \left( \frac{P \cdot Q}{q \cdot \widetilde{P} \cdot Q + (1-q) \cdot P \cdot \widetilde{Q}} \right)^{\alpha} \big( q \cdot \widetilde{P} \cdot Q + (1-q) \cdot P \cdot \widetilde{Q} \big) \\
= & \left( \frac{P \cdot Q}{q \cdot \widetilde{P} \cdot Q + (1-q) \cdot P \cdot \widetilde{Q}} \right)^{\alpha - 1} \cdot P \cdot Q \\
= & \left( q \cdot \frac{\widetilde{P}}{P} + (1-q) \cdot \frac{\widetilde{Q}}{Q} \right)^{1 - \alpha} \cdot P \cdot Q \\
= & \left( q \cdot \frac{\widetilde{P}}{P} \frac{Q}{\widetilde{Q}} + (1-q) \right)^{1 - \alpha} \cdot \left( \frac{\widetilde{Q}}{Q} \right)^{1 - \alpha} \cdot P \cdot Q \\
= & \left( q \cdot \frac{\widetilde{P}}{P} \frac{Q}{\widetilde{Q}} + (1-q) \right)^{1 - \alpha} \cdot \left( \frac{Q}{\widetilde{Q}} \right)^{\alpha - 1} \cdot P \cdot Q \\
\le & \left( q \cdot \frac{P}{\widetilde{P}} \frac{\widetilde{Q}}{Q} + (1-q) \right)^{\alpha - 1} \cdot \left( \frac{Q}{\widetilde{Q}} \right)^{\alpha - 1} \cdot P \cdot Q \\
= & \left( q \cdot \frac{P}{\widetilde{P}} \frac{\widetilde{Q}}{Q} + (1-q) \right)^{\alpha - 1} \cdot \left( \frac{Q}{\widetilde{Q}} \right)^{\alpha} \cdot P \cdot \widetilde{Q},
\end{aligned}
\tag{C.9}
$$

where in the inequality we have used Lemma 10. Now we can expand $\left(q \cdot \frac{P}{\widetilde{P}} \frac{\widetilde{Q}}{Q} + 1 - q\right)^{\alpha-1}$:

$$\left(1 - q + q \cdot \frac{P}{\widetilde{P}} \frac{\widetilde{Q}}{Q}\right)^{\alpha-1} \cdot \left(\frac{Q}{\widetilde{Q}}\right)^{\alpha} \cdot P \cdot \widetilde{Q}$$

$$= \left(\sum_{j=0}^{\alpha-1} \binom{\alpha-1}{j} q^j \cdot (1-q)^{\alpha-1-j} \cdot \left(\frac{P}{\widetilde{P}}\right)^j \left(\frac{\widetilde{Q}}{Q}\right)^j\right) \cdot \left(\frac{Q}{\widetilde{Q}}\right)^{\alpha} \cdot P \cdot \widetilde{Q}$$

$$= \left(\sum_{j=0}^{\alpha-1} \binom{\alpha-1}{j} q^j \cdot (1-q)^{\alpha-1-j} \cdot \left(\frac{P}{\widetilde{P}}\right)^j \left(\frac{Q}{\widetilde{Q}}\right)^{\alpha-j}\right) \cdot P \cdot \widetilde{Q}$$

$$= \sum_{j=0}^{\alpha-1} \binom{\alpha-1}{j} q^j \cdot (1-q)^{\alpha-1-j} \cdot \left(\frac{P}{\widetilde{P}}\right)^{j+1} \widetilde{P} \cdot \left(\frac{Q}{\widetilde{Q}}\right)^{\alpha-j} \widetilde{Q}.$$

Then, we use the known $\varepsilon_1(\alpha)$ and $\varepsilon_2(\alpha)$-values as in the inequalities (C.8) to arrive at the claim.

$\square$

## C.1 Auxiliary Lemma

We need the following inequality for the proof of Thm 6.

**Lemma 10** (Lemma 35, Steinke 2022). *For all $p \in [0, 1]$ and $x \in (0, \infty)$,*

$$\frac{1}{1 - p + \frac{p}{x}} \leq 1 - p + p \cdot x.$$

# D Additional Details to Section 4

## D.1 Random Search

Here, we assume we are given some distributions of the hyperparameter candidates and the algorithm $Q$ draws hyperparameters using them. In order to adjust the noise level for each candidate, we take a $\alpha$-line as an RDP upper bound. More specifically, we require that the candidate models are $(\alpha, c \cdot \alpha)$-RDP for some $c > 0$ and for all $\alpha \in \Lambda$. Then the noise scale $\sigma_{\gamma,T}$ for each draw of $(\gamma, T)$ is the minimum scale for which the $(\alpha, c \cdot \alpha)$-RDP bound holds, i.e.,

$$\sigma_{\gamma,T} = \min\{\sigma \in \mathbb{R}^+ \ : \ T \cdot \varepsilon_{\gamma,\sigma}(\alpha) \leq c \cdot \alpha \text{ for all } \alpha \in \Lambda\}.$$

Similarly, we can find the maximum $T$ based on $\sigma$ and $\gamma$ such that the mechanism is $(\alpha, c \cdot \alpha)$-RDP for all $\alpha \in \Lambda$. Again, by Lemma 9 below, the candidate picking algorithm $Q$ is then $c \cdot \alpha$-RDP and we may use Thm. 5 to obtain RDP bounds for the tuning algorithm.

### D.1.1 Adjusting the Parameters $T$ and $\sigma$ for DP-SGD

We next discuss the reasons for the success of strategies described in Section 4. It is often a good approximation to say that the RDP-guarantees of the Poisson subsampled Gaussian mechanism are lines as functions of the RDP order $\alpha$, i.e., that the guarantees are those a Gaussian mechanism with some sensitivity and noise level values. For example, (Thm. 11, Mironov et al., 2019) show that the Poisson subsampled Gaussian mechanism is $(\alpha, 2\gamma^2\alpha/\sigma^2)$-RDP when $\alpha$ is sufficiently small. Also, (Thm. 38, Steinke, 2022) show that if the underlying mechanism is $\rho$-zCDP, then the Poisson subsampled version with subsampling ratio $\gamma$ is $(\alpha, 10\gamma^2\rho\alpha)$-RDP when $\alpha$ is sufficiently small. Notice that the Gaussian mechanism with $L_2$-sensitivity $\Delta$ and noise level $\sigma$ is $(\Delta^2/2\sigma^2)$-zCDP (Bun and Steinke, 2016).

We numerically observe, that the larger the noise level $\sigma$ and the smaller the subsampling ratio $\gamma$, the better the line approximation of the RDP-guarantees (see Figure 9).

In case the privacy guarantees (either $(\varepsilon, \delta)$-DP or RDP) are approximately those of a Gaussian mechanisms with some sensitivity and noise level values, both of the approaches for tuning the hyperparameters $\gamma$, $\sigma$ and $T$ described in Section 4 would lead to very little slack. This is because for the Gaussian mechanism, both the RDP guarantees (Mironov, 2017) and $(\varepsilon, \delta)$-DP guarantees (Dong et al., 2022) depend monotonously on the scaled parameter

$$\widetilde{\sigma} = \frac{\sigma}{\Delta \cdot \sqrt{T}}.$$

This means that if we adjust the training length $T$ based on values of $\sigma$ by having some target $(\delta, \varepsilon)$-bound for the candidate model with grid search of Section 4.1, the resulting RDP upper bounds of different candidates will not be far from each other (and similarly for adjusting $\sigma$ based on value of $T$). Similarly, for random search of Section D.1, when adjusting $T$ based on values of $\sigma$, the RDP guarantees of all the candidate models would be close to the upper bound ($c \cdot \alpha$, $c > 0$), i.e., they would not be far from each other.

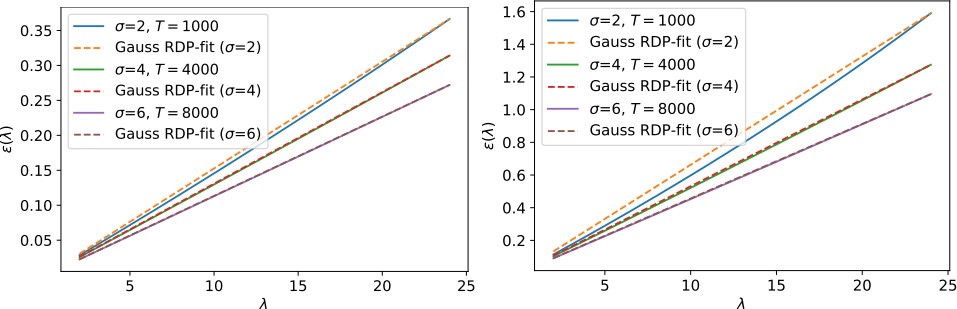

Figure 9: DP-SGD RDP curves for different values of noise level $\sigma$ and number of compostions $T$. Left: $\gamma = 1/100$, right: $\gamma = 1/50$ and the corresponding lines with the smallest slope that give upper bounds for the RDP orders up to $\alpha = 24$.

## D.2 Proof of Lemma 9

**Lemma 11.** *Denote by $\beta$ the random variable of which outcomes are the hyperparameter candidates (drawing either randomly from a grid or from given distributions). Consider an algorithm $Q$, that first randomly picks hyperparameters $t \sim \beta$, then runs a randomised mechanism $\mathcal{M}(t, X)$. Suppose $\mathcal{M}(t, X)$ is $\big(\alpha, \varepsilon(\alpha)\big)$-RDP for all $t$. Then, $Q$ is $\big(\alpha, \varepsilon(\alpha)\big)$-RDP.*

*Proof.* Suppose the hyperparameters $t$ are outcomes of a random variable $\beta$. Let $X$ and $Y$ be neighbouring datasets. Then, if $p(t, s)$ and $q(t, s)$ (as functions of $s$) give the density functions of $\mathcal{M}(t, X)$ and $\mathcal{M}(t, Y)$, respectively, we have that

$$Q(X) \sim \mathbb{E}_{t \sim \beta}\, p(t, s) \quad \text{and} \quad Q(Y) \sim \mathbb{E}_{t \sim \beta}\, q(t, s).$$

Since for any distributions $p$ and $q$, and for any $\alpha > 1$, $\exp\big((\alpha - 1)D_\alpha(p\|q)\big) = \int \left(\frac{p(t)}{q(t)}\right)^\alpha q(t)\, \mathrm{d}t$ gives an $f$-divergence (for $f(x) = x^\alpha$), it is also jointly convex w.r.t. $p$ and $q$ (Liese and Vajda, 2006). Using Jensen's inequality, we have

$$\exp\big((\alpha - 1)D_\alpha\big(Q(X)\|Q(Y)\big)\big) = \int \left(\frac{\mathbb{E}_{t \sim \beta}\, p(t, s)}{\mathbb{E}_{t \sim \beta}\, q(t, s)}\right)^\alpha \cdot \mathbb{E}_{t \sim \beta}\, q(t, s)\, \mathrm{d}s$$

$$\leq \mathbb{E}_{t \sim \beta} \int \left(\frac{p(t, s)}{q(t, s)}\right)^\alpha \cdot q(t, s)\, \mathrm{d}s$$

$$\leq \mathbb{E}_{t \sim \beta} \exp\big((\alpha - 1)D_\alpha\big(\mathcal{M}(t, X)\|\mathcal{M}(t, Y)\big)\big)$$

$$= \exp\big((\alpha - 1)\varepsilon(\alpha)\big)$$

from which the claim follows. □

# E  $f$-Divergence of Parallel Compositions

We first formulate the parallel composition result for general $f$-divergences (Lemma 12). We then obtain the RDP bound for parallel compositions as a corollary (Cor. 13).

Our Lemma 12 below can be seen as an $f$-divergence version of the $(\varepsilon, 0)$-DP result given in (Thm. 4 McSherry, 2009). Corollary 2 by (Smith et al., 2022) gives the corresponding result in terms of $\mu$-Gaussian differential privacy (GDP), and it is a special case of our Lemma 12 as $\mu$-GDP equals the $(\varepsilon, \delta)$-DP (i.e., the hockey-stick divergence) of the Gaussian mechanism with a certain noise scale (Cor. 1, Dong et al., 2022).

We define $f$-divergence for distributions on $\mathbb{R}^d$ as follows. Consider two probability densities $P$ and $Q$ defined on $\mathbb{R}^d$, such that if $Q(x) = 0$ then also $P(x) = 0$, and a convex function $f : [0, \infty) \to \mathbb{R}$. Then, an $f$-divergence (Liese and Vajda, 2006) is defined as

$$D_f(P\|Q) = \int f\left(\frac{P(t)}{Q(t)}\right) Q(t) \, \mathrm{d}t.$$

In case the data is divided into disjoint shards and separate mechanisms are applied to each shard, the $f$-divergence upper bound for two neighbouring datasets can be obtained from the individual $f$-divergence upper bounds:

**Lemma 12.** *Suppose a dataset $X \in \mathcal{X}^N$ is divided into $k$ disjoint shards $X_i$, $i \in [k]$, and mechanisms $\mathcal{M}_i$, $i \in [k]$, are applied to the shards, respectively. Consider the adaptive composition*

$$\mathcal{M}(X) = \big(\mathcal{M}_1(X_1), \mathcal{M}_2(X_2, \mathcal{M}_1(X_1)), \ldots, \mathcal{M}_k(X_k, \mathcal{M}_1(X_1), \ldots, \mathcal{M}_{k-1}(X_{k-1}))\big).$$

*Then, we have that*

$$\max_{X \sim Y} D_f\big(\mathcal{M}(X)\|\mathcal{M}(Y)\big) \leq \max_{i \in [k]} \max_{X \sim Y} D_f\big(\mathcal{M}_i(X)\|\mathcal{M}_i(Y)\big).$$

*Proof.* Let $X$ and $Y$ be divided into $k$ equal-sized disjoint shards and suppose $Y$ is a neighbouring dataset such that $X$ and $Y$ differ in $j^{th}$ shard, i.e., $X_j \sim Y_j$ and $Y = (Y_1, Y_2, \ldots, Y_k) = (X_1, \ldots, X_{j-1}, Y_j, X_{j+1}, \ldots, X_k)$.

Then, we see that

$$
\frac{\mathbb{P}\big(\mathcal{M}(X) = (a_1, \ldots, a_k)\big)}{\mathbb{P}\big(\mathcal{M}(Y) = (a_1, \ldots, a_k)\big)}
$$

$$
= \frac{\mathbb{P}\big(\mathcal{M}_1(X_1) = a_1\big) \cdot \mathbb{P}\big(\mathcal{M}_1(X_2, a_1) = a_2\big) \cdots \mathbb{P}\big(\mathcal{M}_k(X_k, a_1, \ldots, a_{k-1}) = a_k\big)}{\mathbb{P}\big(\mathcal{M}_1(Y_1) = a_1\big) \cdot \mathbb{P}\big(\mathcal{M}_1(Y_2, a_1) = a_2\big) \cdots \mathbb{P}\big(\mathcal{M}_k(Y_k, a_1, \ldots, a_{k-1}) = a_k\big)}
$$

$$
= \frac{\mathbb{P}\big(\mathcal{M}_j(X_j, a_1, \ldots, a_{j-1}) = a_j\big)}{\mathbb{P}\big(\mathcal{M}_j(Y_j, a_1, \ldots, a_{j-1}) = a_j\big)}.
$$

and furthermore, denoting $a = (a_1, \ldots, a_k)$,

$$D_f\big(\mathcal{M}(X)\|\mathcal{M}(Y)\big))$$

$$= \int f\left(\frac{\mathbb{P}\big(\mathcal{M}(X) = a\big)}{\mathbb{P}\big(\mathcal{M}(Y) = a\big)}\right) \mathbb{P}\big(\mathcal{M}(Y) = a\big) \, \mathrm{d}a$$

$$= \int f\left(\frac{\mathbb{P}\big(\mathcal{M}_j(X_j, a_1, \ldots, a_{j-1}) = a_j\big)}{\mathbb{P}\big(\mathcal{M}_j(Y_j, a_1, \ldots, a_{j-1}) = a_j\big)}\right) \mathbb{P}\big(\mathcal{M}(Y) = a\big) \, \mathrm{d}a$$

$$= \int f\left(\frac{\mathbb{P}\big(\mathcal{M}_j(X_j, a_1, \ldots, a_{j-1}) = a_j\big)}{\mathbb{P}\big(\mathcal{M}_j(Y_j, a_1, \ldots, a_{j-1}) = a_j\big)}\right) \mathbb{P}\big(\mathcal{M}_1(Y_1) = a_1\big) \cdot \mathbb{P}\big(\mathcal{M}_1(Y_2, a_1) = a_2\big) \cdots$$

$$\mathbb{P}\big(\mathcal{M}_j(Y_j, a_1, \ldots, a_{j-1}) = a_j\big) \, \mathrm{d}a_1 \ldots \mathrm{d}a_j$$

$$= D_f(\mathcal{M}_j(X_j)\|\mathcal{M}_j(Y_j))) \cdot \int \mathbb{P}\big(\mathcal{M}_1(Y_1) = a_1\big) \cdot \mathbb{P}\big(\mathcal{M}_1(Y_2, a_1) = a_2\big) \cdots$$

$$\mathbb{P}\big(\mathcal{M}_{j-1}(Y_{j-1}, a_1, \ldots, a_{j-2}) = a_{j-1}\big) \, \mathrm{d}a_1 \ldots \mathrm{d}a_{j-1}$$

$$= D_f(\mathcal{M}_j(X_j)\|\mathcal{M}_j(Y_j))).$$

Thus,

$$D_f(\mathcal{M}(X)||\mathcal{M}(Y)) = D_f(\mathcal{M}_j(X_j)||\mathcal{M}_j(Y_j)) \leq \max_{X \sim Y} D_f(\mathcal{M}_j(X)||\mathcal{M}_j(Y))$$

and also, we have that

$$\max_{X \sim Y} D_f(\mathcal{M}(X)||\mathcal{M}(Y)) = \max_{i \in [k]} \max_{X \sim Y} D_f(\mathcal{M}_i(X)||\mathcal{M}_i(Y)).$$

$\square$

**Corollary 13.** *Suppose a dataset $X \in \mathcal{X}^N$ is divided into $k$ disjoint shards $X_i$, $i \in [k]$, and mechanisms $\mathcal{M}_i$, $i \in [k]$, are applied to the shards, respectively. Consider the mechanism*

$$\mathcal{M}(X) = \big(\mathcal{M}_1(X_1), \ldots, \mathcal{M}_k(X_k)\big).$$

*Suppose each $\mathcal{M}_i$ is $\big(\alpha, \varepsilon_i(\alpha)\big)$-RDP, respectively. Then, $\mathcal{M}$ is $\big(\alpha, \max_{i \in [k]} \varepsilon_i(\alpha)\big)$-RDP.*

*Proof.* This follows from Lemma 12 since

$$\exp\big((\alpha-1)D_\alpha(\mathcal{M}(X)||\mathcal{M}(Y))\big) = \int \left(\frac{\mathbb{P}\big(\mathcal{M}(X)=a\big)}{\mathbb{P}\big(\mathcal{M}(Y)=a\big)}\right)^\alpha \mathbb{P}\big(\mathcal{M}(Y)=a\big)\ \mathrm{d}a$$

is an $f$-divergence for $f(x) = x^\alpha$. Thus, by Lemma 12 we have that

$$\max_{X \sim Y} \exp\big((\alpha-1)D_\alpha(\mathcal{M}(X)||\mathcal{M}(Y))\big) \leq \max_{i \in [k]} \max_{X \sim Y} \exp\big((\alpha-1)D_\alpha(\mathcal{M}_i(X)||\mathcal{M}_i(Y))\big)$$

from which it follows that

$$\max_{X \sim Y} D_\alpha(\mathcal{M}(X)||\mathcal{M}(Y)) \leq \max_{i \in [k]} \max_{X \sim Y} D_\alpha(\mathcal{M}_i(X)||\mathcal{M}_i(Y)) = \max_{i \in [k]} \varepsilon_i(\alpha).$$

$\square$

