# OpenReview forum: "Practical Differentially Private Hyperparameter Tuning with Subsampling"
_NeurIPS.cc/2023/Conference — NeurIPS 2023 poster_

### Official Review · Reviewer_kuHB · 2023-07-04

**Soundness:** 4 excellent
**Presentation:** 3 good
**Contribution:** 3 good
**Rating:** 6
**Confidence:** 4

**Summary:**

The paper focuses on the problem of differentially private (DP) hyperparameter tuning for DP ML algorithms. The authors propose a method that subsamples the sensitive data for hyperparameter tuning, and extrapolates optimal values from the small dataset to the larger dataset. The authors provide a Renyi DP (RDP) analysis for their method, show that their method lowers the DP bound and computational complexity of hyperparameter tuning, and empirically show their method leads to better privacy-utility trade-offs than the existing SOTA method.


**Strengths:**

1. The paper focuses on improving private hyperparameter tuning from all facets: using subsampled inputs to save on privacy cost, using the model output by the tuned choices for initializing the training over the larger dataset, as well as extrapolating optimal values for various hyperparameters from the smaller dataset to the larger dataset.
2. The paper is well written.


**Weaknesses:**

Though the authors discuss this in Section 6, I feel one of the main shortcomings of the proposed method is the extrapolation heuristics designed specifically for DP-SGD. Having the method being applicable to commonly used adaptive optimizers (also discussed in Section 6) would make it more general.

**Questions:**

I have listed the main limitation I felt in the weaknesses section.


------------
Post-rebuttal update: It is great to see the authors being able to provide guidance on DP-Adam using their tuning! I am raising my score since this addresses my main concern for the paper.

**Limitations:**

Yes.

---

> ### Author Rebuttal · Authors · 2023-08-09
>
> > Though the authors discuss this in Section 6, I feel one of the main shortcomings of the proposed method is the extrapolation heuristics designed specifically for DP-SGD. Having the method being applicable to commonly used adaptive optimizers (also discussed in Section 6) would make it more general.
>
> We believe our method is applicable also to other optimizers, however the scaling laws may need to be considered case by case for each optimizer. In the attached pdf file we show positive results for Adam with DP gradients. We use the same scaling of the hyperparameters as for DP-SGD except that we keep the learning rate constant when transferring the optimal hyperparameters from the tuning dataset to the larger dataset. This can be heuristically motivated by the non-DP case. For example, the original Adam paper by Kingma and Ba (2015) (https://arxiv.org/pdf/2305.13209.pdf) recommends using the same initial learning rate $\alpha=0.001$ for all (non-DP) ML model training. We experimentally found that the DP-SGD scaling where we scale the learning rate proportionally to the dataset size works poorly for DP-Adam. We plan to add the DP-Adam results to the appendix.
>
> Lastly, we think that building a working hyperparameter tuning framework for DP-SGD which cuts the privacy costs and saves computations is already sufficient for a single paper. More generally, we think any practical and well-motivated solution is a worthwhile contribution on its own even if it would be limited to DP-SGD.

---

> ### Author Response · Authors · 2023-08-17
>
> Thanks a lot for considering our response. We will take your feedback into account when revising.

---

### Official Review · Reviewer_on32 · 2023-07-06

**Soundness:** 3 good
**Presentation:** 3 good
**Contribution:** 4 excellent
**Rating:** 6
**Confidence:** 3

**Summary:**

This paper is about the efficient tuning of hyperparameters of machine learning models in a differentially private (DP) way. Previous works have shown that hyperparameter optimization can be computationally demanding and, more importantly, often uses sensitive data that might get leaked. However, this leakage can be reduced. Specifically in this paper, the authors aim for lowering not only the DP bounds but also the computational demand. The latter is achieved by only using a random subset of data for the hyperparameter optimization. As in previous work, a Rényi DP analysis is provided. Experiments under various settings and data sets demonstrate a better privacy-performance trade-off than baseline competitors while also being faster.

**Strengths:**

- the problem of having efficient differentially private strategies for hyperparameter optimization is relevant and important
- using subsampling to improve the computational efficiency and using its privacy amplifying effect is natural
- the paper is mostly easy to read and follow (there are some issues, see below)
- the theoretical analysis and experimental evaluation shows that either a better performance for the same privacy budget or a better privacy budget for the same performance can be achieved
- the approach is computationally less demanding than previous works

**Weaknesses:**

Overall, the paper appears to be solid, however, I think the paper lacks (a bit of) clarity. Some concepts (e.g., extrapolating hyperparameters) are insufficiently or too late introduced and other choices such as the training on all data but the previously used subset are insufficiently justified. Similar clarity issues appear for the mathematical notation. Moreover, the language should be revised/improved. However, I believe that those issues can be fixed. For almost all points, I provide a list of remarks, suggestions, and questions below.


### Remarks:
- Section 1.1: Using subsets for hyperparameter tuning specifically for deep learning models (in a non-DP setting) was also recently considered, e.g., in:
	- Killamsetty, K., Durga, S., Ramakrishnan, G., De, A., & Iyer, R. (2021, July). Grad-match: Gradient matching based data subset selection for efficient deep model training. In International Conference on Machine Learning (pp. 5464-5474). PMLR.
	- Killamsetty, K., Abhishek, G. S., Lnu, A., Ramakrishnan, G., Evfimievski, A., Popa, L., & Iyer, R. (2022). Automata: Gradient based data subset selection for compute-efficient hyper-parameter tuning. Advances in Neural Information Processing Systems, 35, 28721-28733.
- Definition 2: How is $\mathcal{M}$ an output distribution?
- Theorem 4: Consider mentioning/introducing $\gamma$ for completeness.
- line 106: I consider "number of candidate models $K$" is sub-optimally phrased. A hyperparameter can also be the learning rate, which does not change the model, but the optimization/learning of it. Perhaps something like "the number of runs $K$ within the hyperparemeter tuning" is clearer? This related to my first minor remark (lines 4-5).
- Theorem 5: The notation $\mathcal{X}^N$ was used in Papernot and Steinke but the superscript $N$ was not introduced in this paper. Besides, $n$ instead of $N$ was used. Moreover, $\mathcal{Y}$ denotes the outputs in Papernot and Steinke whereas in this paper, $\mathcal{O}$ was used!
- line 128: How is $\mathcal{M}_1$ different from $A$ in Theorem 5? Both are "hyperparameter tuning algorithms".
- line 130: Aren't $\theta_1$ the model *parameters* instead of the model itself? In Equation (2.3), $f$ is the model with parameters $\theta$.
- Equation (3.1): $\mathcal{M}_1(X_1)$ returns a tuple but in Equation (3.1) it is assumed that it only returns the second entry of that tuple. Besides, the output of $\mathcal{M}(X)$ would then be $((\theta_1, t_1), \theta_2)$.
- Theorem 6: The notation $\mathcal{X}^n$ was never introduced. Besides, if a data set $X$ is an element of $\mathcal{X}$, then a single data point $x$ cannot be. Instead, it should be $x$ in curly brackets, no?
- Lemma 9: First, $C$ was used as a scalar within the clipping strategy. Hence, the variable name cannot be used anymore for something else. Second, $\mathcal{M}$ never took $t$ as an argument before. Consider adapting the notation.



### Minor remarks and suggestions:
- lines 4-5, 35-36: I think "where the number of random search samples is randomized itself" could be stated more clearly.
- line 21, 96, 105, 107, 253, 280: however,
- line 34: "provided a" instead of "gave" (there should be an a)
- lines 53, 56, 225: w.r.t. the number
- line 54: "than" instead of "then"
- line 56: In a non-DP
- line 74: curly brackets for sets
- line 85: "As is common in practice,"
- line 87 (and others): Equation (2.2)
- line 90: to have an $L_2$-norm of at most
- Equation (2.3): Consider mentioning $f$ and $\theta$ for completeness. You could also connect $Z_j$ with the noise mentioned in line 90.
- line 95, 106, 171: Wrong citation style (direct instead of indirect, i.e., by Zhu and Wang, (2019)).
- line 97: If you want to gain space you can try adding `\looseness=-1` after "method." which tells $\LaTeX$ to try to fit "method." into the paragraph. Might also work for the caption of Figure 3. It is a neat trick worth knowing.
- line 124: Inconsistent capitalization: Method should be capitalized
- line 129: the method by
- line 134: either use or consider
- Consistency: "dataset" vs "data set" (e.g., lines 131 vs 134)
- line 135: Math (Equation (3.2)) is always a part of a sentence. Hence, use rather a colon instead of a full-stop.
- line 136: , respectively
- line 137: with a standard [...] Theorem 4
- line 142: The number of iterations $T$ appear for the first time. Consider introducing them earlier, i.e., around lines 89-97.
- line 165: Adam is an ICLR paper, not a preprint.
- line 169: the learning rate
- Consistency: "Thm." vs "Theorem" (e.g., lines 173 vs 212)
- line 193-194: methods [...] as functions
- line 194: mechanism DP-SGD is run
- It is slightly confusing to mix "variants 1 and 2" and "(3.1) and (3.2)".
- Figure 1: the three dashed lines between "our method" and "variant" might be confusing in black and white prints since they look like the blue curve. The crossing of lines around $q=0.1$ could be highlighted with an arrow.
- Consistency: $E$ vs E in lines 202-204.
- [Section 4.1 (but also before): It is slightly weird that some lines are missing line numbers.]
- Section 5: Please name briefly at least the data sets and the learning task. Space can be gained be vertically shrinking Figures 2 and 3.
- line 245: on the
- line 247: 1150 and 1278, respectively.
- line 251: Next,
- line 254: typo in hyperparameter
- Figure 2: What about "$6.06 \times$ slower"? Perhaps that is more clear/less confusing.
- Figure 2 caption: Figure 4 (appendix) shows a more detailed version of this figure.
- line 273: e.g. already means for example
- line 276: "question is" or "questions are"
- line 280: remove on
- Tables: Captions go above tables and vertical lines should be avoided, see https://media.neurips.cc/Conferences/NeurIPS2023/Styles/neurips_2023.pdf

### After the Rebuttal:
I have read all other reviews and all rebuttals. Furthermore, I thank the authors for their answers and the additional insights provided in the extra rabuttal pdf. I keep my original score.

**Questions:**

- Q1: What is meant by "extrapolate the hyperparameter values to a larger dataset"? Can you provide an example? It is mentioned multiple times (lines 10, 43, 123, 131, ...) and I think it should be clarified (earlier). If the main example is in lines 140-146, I think it might help to tease it earlier as I was unsure what might be meant by extrapolating. Additionally, is there another example of extrapolation of hyperparameters?
- Q2: Why is $X_1$ being excluded from $X$ in step 4 in line 132? I see the second variant as being meaningful but what is the motivation/intuition/reasoning behind variant 1?
- Q3: Regarding the adjustment of the learning rate (lines 140-146): Didn't van der Veen et al. (2018) consider a change in batch size with a fix data set size? If the same batch size is used for the subsampled data set and the data set, it should not matter, or?
- Q4: I am slightly confused by the difference of $\gamma$ and $q$, i.e., in Theorem 4 (and lines 193-197). My understanding is that $\gamma$ is for the batch and $q$ is for the subset $X_1$ (line 126). Can you clarify the difference between $\gamma$ and $q$?
- Q5: Can you elaborate on how Figure 1 was created? As in, where do the numbers come from and how do the specific choices (title of the subfigures) go into that?
- Q6: Can you elaborate on what is shown in Figures 2 and 3? Is the complete hyperparameter optimization cycle run ten times and the figures show the averaged test accuracies of the final models (i.e., the ten models with the best hyperparameters)? What are the individual dots per line?
- Q7: The choice of $\mu$ is fixed for Figure 2 but quite differs in Figure 3. CIFAR-10 is harder than MNIST but has less "candidate models". How did you come up with those $\mu$ values? Does CIFAR-10 has the smallest $\mu$ because it uses a pre-trained model and all others are trained from scratch?
- Q8: How is the choice of Adam (DP-SGD gradients) justified when the results only hold for DP-SGD? Did IMDB not work with DP-SGD? What was the reason for this choice?
- Q9: Can you provide a table with data set sizes (and dimensionalities) as well as the used subset sizes?
- Q10: Regarding the computational savings it is stated that the number in the plots next to baseline refers to the mean training time of the baseline scaled wrt the minimum of the variants. Can you provide more insight in those computational savings? Is it just the worst performing model on the smallest subset that yielded this saving once? Are your variants sometimes even slower than the baseline? Is there a substantial difference between the variants?

**Limitations:**

Limitations and potential negative societal impact are discussed.

---

> ### Author Rebuttal · Authors · 2023-08-04
>
>
> We thank the reviewer for the careful reading. Addressing these comments will definitely improve the paper. Thank you also for the references on related non-DP methods that went unnoticed, we will add them to the literature section. Below we answer the questions, we will address the minor remarks and suggestions in the possible final version of the paper.
>
> > Q1: What is meant by "extrapolate the hyperparameter values to a larger dataset"?
>
> Thank you for pointing out this. We are thinking of the optimal hyperparameters as a function of the dataset size, and by finding the optimal values for a small dataset, we can determine the hyperparameter values for the larger dataset using our scaling rules (Section 3.2). We admit that this is a bit vaguely expressed and we can change the word ‘extrapolating’ to ‘transferring’. For example on line 10 we could say instead: "… and by transferring the optimal values to a larger dataset.”, on line 131 “ Transfer the hyperparameters $t_1$ to the dataset…”.
>
> > Q2:  .. what is the motivation/intuition/reasoning behind variant 1?
>
> By dedicating $X_1$ for the tuning part, the overall privacy costs becomes smaller since $X_1$ does not participate on the final model training (see Fig. 1). We also have in mind some practical scenarios, where there could be a data owner that would only report the optimal hyperparameters. Although not reflected by the DP guarantees, one could claim that the attack surface of the hyperparameter values is small compared to the model parameters and thus the tuning data would have a different kind of protection. One could think of, e.g., a federated learning scenario, where this could be utilized.
>
> > Q3: Didn't van der Veen et al. (2018) consider a change in batch size with a fix data set size?
>
> You are right, van der Veen et al. scale with the batch size. We will replace “dataset size” with “batch size” on line 145. In our case however these are equivalent, since in order to have the same privacy guarantees for the candidate models trained using the small dataset and using the large dataset, we need to keep the subsampling ratio constant (which means scaling the batch size).
>
> >Q4: ... My understanding is that $\gamma$ is for the batch and $q$ is for the subset $X_1$ (line 126).
>
> Indeed, $\gamma$ is for the mini batches and $q$ is for the subset $X_1$. There is a typo in statement of Thm. 4: $q$ should be $\gamma$. We will add a short clarification before Thm. 4 on the meaning of $\gamma$.
>
> > Q5: Can you elaborate on how Figure 1 was created? As in, where do the numbers come from and how do the specific choices (title of the subfigures) go into that?
>
> These are simply some commonly used parameter values for DP-SGD (see, e.g., the experiments of Abadi et al., 2016) used to illustrate the differences between the various DP bounds. Notice that increasing $\mu$ increases the privacy cost of the tuning part.
>
> >Q6 (Figures 2 and 3) Is the complete hyperparameter optimization cycle run ten times and the figures show the averaged test accuracies of the final models ... ?
>
> Yes, exactly, with 10 different seeds.
>
> >What are the individual dots per line?
>
> Each dot corresponds to different overall $(\varepsilon,\delta)$-bound we predefine. In case of learning rate tuning (Figure 2) we keep the subsampling ratio $\gamma$ of DP-SGD constant and the $\sigma$ will be determined by the $\varepsilon$-bound ($\delta$ is fixed to $10^{-5}$ as described in the appendix). In Figure 3, where we tune also $\gamma$ and number of epochs, $\sigma$ is again determined by the $\varepsilon$-bound. We will clarify this more carefully.
>
> > Q7: How did you come up with those $\mu$ values? Does CIFAR-10 has the smallest because it uses a pre-trained model and all others are trained from scratch?
>
> Yes, that is roughly the reasoning. Pretrained CIFAR-10 is a fairly small convex model (only the last layer trained). Fashion MNIST has the most number of hyperparameter choices (80 combinations in total, see Appendix Table 2) so it has slightly larger $\mu$ than IMDB and MNIST (54 and 72 combinations, respectively). However, we believe we would observe similar improvements also when fixing $\mu$ values, as the baseline method uses the same $\mu$ values as well.
>
> > Q8: How is the choice of Adam (DP-SGD gradients) justified when the results only hold for DP-SGD? Did IMDB not work with DP-SGD? What was the reason for this choice?
>
> This was choice was dictated by the Opacus repository recommendation: Adam with DP gradients seems to work better than DP-SDG for IMDB (https://github.com/pytorch/opacus/blob/main/examples/imdb_README.md). We observed that tuned DP-SGD was better using our approach than using the baseline tuning method (see the first figure in the attached pdf file).
>
> > Q9: Can you provide a table with data set sizes (and dimensionalities) as well as the used subset sizes?
>
> Yes, we will add this in the appendix.
>
> > Q10: Are your variants sometimes even slower than the baseline? Is there a substantial difference between the variants?
>
> In expectation our variants are much faster than the baseline. The main speedup comes from the fact that in the tuning procedure our methods run in expectation the same number of candidate models as the baseline, however with much less data. For same combination of hyperparameters values, for training a candidate model, our methods run the same number of iterations as the baseline, however with much smaller batches (since the subsampling ratio is kept constant). E.g., if the tuning set $X_1$ is 10 times smaller than the whole data $X$, there will be in expectation 10 times fewer gradient evaluations in the tuning procedure, compared to the baseline. Our variant 1 is slightly faster than variant 2, since the final model training is carried out with less data, however the overall difference is small.

---

> > ### Comment · Reviewer_on32 · 2023-08-14
> >
> > Thank you very much for your answers and the additional insights provided in the extra rabuttal pdf. It definitely added clarity for me. I appreciate it.

---

> > > ### Author Response · Authors · 2023-08-15
> > >
> > > Thank you once again for the helpful review. We will take into account these remarks and suggestions in the revised version of the paper.

---

### Official Review · Reviewer_zJd6 · 2023-07-06

**Soundness:** 4 excellent
**Presentation:** 3 good
**Contribution:** 3 good
**Rating:** 7
**Confidence:** 4

**Summary:**

This work proposes a practical variant of tuning hyperparameters under differential privacy (DP) which has a better privacy-utility tradeoff and less computational time compared to the SOTA private hyperparameter tuning algorithm. Instead of using all datasets to tune hyperparameters as in previous works, this work proposes to use a subsampled dataset for hyperparameter tuning and provides a Renyi DP (RDP) analysis of the proposed approach. In addition, this work considers tuning hyperparameters that can affect the privacy loss. Both theory and experimental results demonstrate the superior performance of the proposed approach in terms of the utility-privacy tradeoff and computation time.

**Strengths:**

Clear and well-structured presentation.

Simple yet effective approach.

Interesting results.


**Weaknesses:**

There are a few confusing points. See questions below.

**Questions:**

There is no description of the subsampling ratio $\gamma$. Is $\gamma$ the subsampling ratio for selecting minibatches in DP-SGD?

Regarding figure 1:

-	Are the blue curves generated by taking max{Eq.(3.6), Eq.(3.7)} in Theorem 6?
-	What is the alpha used here?
-	Since the caption says “($\epsilon$, $\delta$)”-bounds, is it that the privacy loss of the proposed method is stated in RDP, and when generating the plots for comparison, the RDP bound is converted to a DP bound?
-	Is the $\delta$ part in the DP bound comparable?

What is $\Delta W$ in Eq.(3.3)?

Line 227: What is $\Lambda$ (the set containing all alpha values) used in the experiments?

What are the major challenges of analyzing an extended version of this approach to tune the hyperparameters of other private optimizers, e.g., DP-Adam?

Minor comments: when describing the approach (line 126 ~ 132) it might be clearer to use different letters for the base algorithm and the hyperparameter tuning algorithm.

Minor issue: M subsample Poisson(q) should be M subsample Poisson ($\gamma$) in Theorem 4.

Minor issue: it seems the right parentheses is missing in Eq.(3.1), Eq.(3.2).


**Limitations:**

Limitations and broader impact are well discussed in the main paper.

---

> ### Author Rebuttal · Authors · 2023-08-04
>
> Thank you for all these comments and suggestions. These will definitely improve the readability of the paper.
>
> > There is no description of the subsampling ratio $\gamma$. Is the subsampling ratio for selecting minibatches in DP-SGD?
>
> Yes, it is the subsampling ratio for DP-SGD. Its meaning is stated explicitly only on line 141 although we use the notation in the statement of Theorem 4 (line 100). We will add a clarification of the notation before Thm. 4.
>
> > Regarding figure 1:
>
> > Are the blue curves generated by taking max{Eq.(3.6), Eq.(3.7)} in Theorem 6?
>
> Yes. By Def. 2, $\mathcal{M}$ is $\varepsilon(\lambda)$-RDP for max{Eq.(3.6), Eq.(3.7)} . We will add a line in the statement of Thm. 6 where we describe that the mechanism $\mathcal{M}$ is $\varepsilon(\lambda)$-RDP for maximum of the expressions given in Eq.(3.6) and Eq.(3.7). This should make it more readable.
>
> > What is the alpha used here?
>
> We convert the RDP-bounds to $(\varepsilon,\delta)$-bounds using Lemma 3 (Eq. (2.2)), as described on lines 85-87. The statement of Lemma 3 holds for any $\alpha$ so we minimize the expression of Eq. 2.2. w.r.t. $\alpha$ (a common practice). For Figure 1, $\varepsilon(\lambda)$ is always evaluated using our Theorem 6. We will add on line 87: “minimizing over the values given by (2.2) w.r.t. $\alpha$.” and add a short description to lines 193-197 that we use Lemma 2 for the conversion.
>
> > Since the caption says “($\varepsilon,\delta$)”-bounds, is it that the privacy loss of the proposed method is stated in RDP, and when generating the plots for comparison, the RDP bound is converted to a DP bound?
>
> Yes, exactly, we convert using Lemma 2, as described on lines 85-87. We will add a short description to lines 193-197 to clarify this.
>
> > Is the $\delta$ part in the DP bound comparable?
>
> Yes, in all plots of Fig. 1 $\delta$ is fixed to $10^{-5}$, but we notice we have not stated this value. We will add it to lines 193-197. This is only mentioned on line 402 of the Appendix, we will add this also in the experiments section.
>
> > What is $\Delta W$ in Eq.(3.3)?
>
> Here is a typo, there should be $\Delta W \sim \mathcal{N}(0, I_d)$. The same thing in Eq. (3.4).
>
> > Line 227: What is $\Lambda$ (the set containing all alpha values) used in the experiments?
>
> We use integer $\alpha$’s 2 to 64. We will add this at least in the Appendix. Using an integer grid is a common practice in software implementations (e.g. TF privacy) as well. We would be free to choose any grid on $(1,\infty)$, however we do not see noticeable improvement in ($\varepsilon,\delta$)-bounds by enlarging or refining the $\Lambda$ - grid.
>
> > What are the major challenges of analyzing an extended version of this approach to tune the hyperparameters of other private optimizers, e.g., DP-Adam?
>
> One challenge is that we don’t have similar reasoning about the scaling of hyper parameters as we have for DP-SGD (our section 3.2). However, we heuristically find that we get similar improvements when we tune DP-Adam: the scaling of hyperparameters is same as for DP-SGD (Sec. 3.2), except that we keep the learning rate the same when training the final model with the larger dataset. See the attached pdf for experimental comparisons.
>
> > Minor comments: when describing the approach (line 126 ~ 132) it might be clearer to use different letters for the base algorithm and the hyperparameter tuning algorithm.
>
> Thank you, this is a good suggestion, we will try to think of a better notation.
>
> > Minor issue: M subsample Poisson($q$) should be M subsample Poisson ($\gamma$) in Theorem 4.
> > Minor issue: it seems the right parentheses is missing in Eq.(3.1), Eq.(3.2).
>
> Indeed, thank you!

---

> > ### Comment · Reviewer_zJd6 · 2023-08-16
> >
> > Thank you very much for the detailed response!

---

### Official Review · Reviewer_VkdY · 2023-07-09

**Soundness:** 3 good
**Presentation:** 3 good
**Contribution:** 2 fair
**Rating:** 6
**Confidence:** 4

**Summary:**

The paper addresses private hyper-parameter tuning using privacy amplification by sampling. It introduces two versions of private fine-tuning on a sampled subset, which improves upon existing methods that fine-tune the entire private set. The problem involves deriving amplified privacy analysis and scaling the optimal hyper-parameters back to the original dataset. The authors employ linear scaling for hyper-parameter transfer and provide a Renyi differential privacy (RDP) analysis. Empirical results demonstrate that their approaches achieve better trade-offs between privacy and utility.


**Strengths:**

1.  The paper builds upon the existing results of "RDP of hyper-parameter fine-tuning" and "amplification by sampling of RDP" for their privacy analysis. They propose two algorithms for fine-tuning on a sampled set. The first algorithm directly applies the amplification by sampling rule of RDP to the fine-tuning algorithm with a known RDP. The second algorithm involves conducting a hyper-parameter tuning on a subset and then running the mechanism with the selected hyper-parameter on the dataset that does not include the subset.

2. While the contribution of the privacy analysis may appear limited, the authors successfully propose a comprehensive framework capable of tuning both DP and non-DP parameters on a subset. Their empirical results also support their findings regarding the transfer of optimal parameters from the subset to a larger set.

**Weaknesses:**

The contribution of the paper may be considered  incremental as it combines existing two techniques in a straightforward manner, specifically plugging one mechanism's RDP into another amplification by sampling rule that takes a RDP as an input. The privacy analysis of the second algorithm may be considered the only novel theoretical result in the paper.

**Questions:**

The authors propose to use a linear scaling to transfer hyper-parameters. Is it generally hold for all hyper-parameters? For example, shall we also scale the optimal clipping threshold to the dataset size?

---

> ### Author Rebuttal · Authors · 2023-08-09
>
> > The privacy analysis of the second algorithm may be considered the only novel theoretical result in the paper.
>
> We agree that the paper is more on designing a framework for lowering the privacy cost and computational cost of DP-SGD, than on providing new theoretical insights. However, we respectfully disagree with the implicit prerequisite that a DP paper should have a heavy theoretical contribution, especially for a generally applied venue like NeurIPS. As the field has become increasingly applied, there are numerous examples where simple techniques commonly used in non-private settings inspire their DP variants and provide practical utility improvements. While perceived as incremental in retrospect, it is indeed a contribution that our framework gives a much better privacy-utility trade-off and lower computational cost without having to invent a new tuning method/analysis.
>
> Nevertheless, in addition to the RDP analysis, we would like to highlight also our Section 4 (and the related theoretical result of Lemma 9, proof given in Appendix D.2) where we give ways to tune the hyperparameters that affect the RDP guarantees of the candidate models (i.e., when not just tuning the learning rate and clipping constant etc.). In Appendix Section E we state the parallel composition for the Rényi divergence (and for $f$-divergences, more generally) which can be used in case we want to have privacy guarantees, e.g., for the test set as well (we could not find a result suitable for RDP composition in the literature).
>
> > The authors propose to use a linear scaling to transfer hyper-parameters. Is it generally hold for all hyper-parameters? For example, shall we also scale the optimal clipping threshold to the dataset size?
>
> No, this does not hold for all hyperparameters. We explain the DP-SGD scaling in Section 3.2: to keep the privacy costs the same for the candidate models and for the final model, we scale the batch size linearly and keep the noise scale $\sigma$ the same in the transfer. We also keep the clipping constant the same.
>
> As requested by reviewer kuHB we also experimented with an adaptive optimizer, DP-Adam. For DP-Adam we used the same scaling of hyperparameters as for DP-SGD except that we did not scale the learning rate. Our method gives positive results then also (see the attached pdf file).

---

> > ### Comment · Reviewer_VkdY · 2023-08-13
> >
> > Thank you for your response! I agree that ' a general useful framework for hyper-parameter tuning is important to the  community'.  I have raised my score.

---

> > > ### Author Response · Authors · 2023-08-14
> > >
> > > Thank you for taking the time to read our response and taking it into consideration!

---

### Author Rebuttal · Authors · 2023-08-09

We would like to thank all the reviewers for their helpful comments! Addressing them will definitely improve the paper, and our impression is that the required changes are not major. We have replied each review below individually.

We have attached a pdf file with new figures. Here background on them:

Reviewer on32 had a question

> How is the choice of Adam (DP-SGD gradients) justified... Did IMDB not work with DP-SGD? What was the reason for this choice?

Indeed DP-SGD with our tuning method seems to work also for IMDB. The results of the Opacus repository indicated that DP-Adam is the best choice (https://github.com/pytorch/opacus/blob/main/examples/imdb_README.md). However tuned DP-SGD seems to give similar results, and our tuning method is competitive then also. A comparison is given in the pdf.

Reviewer kuHB commented:

> Having the method being applicable to commonly used adaptive optimizers (also discussed in Section 6) would make it more general.

To this end, we have experimented with DP-Adam now on all the four datasets considered, and we notice that we can improve upon the baseline tuning method for DP-Adam as well. The IMDB comparison can already be found in the main text, rest of the DP-Adam comparisons for the learning rate tuning are in the attached pdf.

For DP-Adam we change the learning rate scaling used of DP-SGD (Section 3.2 in our paper) such that we keep the learning rate the same when transferring the optimal hyperparameters to the larger dataset. This can be heuristically motivated by the non-DP case. For example, the original Adam paper by Kingma and Ba (2015) (https://arxiv.org/pdf/2305.13209.pdf) recommends using the same initial learning rate $\alpha=0.001$ for all (non-DP) ML model training. We experimentally found that the DP-SGD learning rate scaling works poorly for DP-Adam. We plan to include the DP-Adam comparisons to the appendix.

---

### Decision · Program_Chairs · 2023-09-21

**Decision:**

Accept (poster)

**Comment:**

This paper proposes methods for making differentially private hyperparameter tuning more efficient. The paper builds on existing tools for differentially private hyperparameter optimization that are not efficient if used directly. To make them efficient, it relies on privacy amplification by sampling, plus heuristics for extrapolating hyperparameter performance on a sample to the full dataset.

This work addresses an important practical problem. It carefully combines various tools to develop a theoretically sound solution and then provides an experimental evaluation. Overall, the reviewers found this submission to be clear and interesting and thus support acceptance at NeurIPS.